# Targeting glutamine-addiction and overcoming CDK4/6 inhibitor resistance in human esophageal squamous cell carcinoma

Shuo Qie[1], Akihiro Yoshida[1], Stuart Parnham [1], Natalia Oleinik[1], Gyda C. Beeson[2], Craig C. Beeson[2], Besim Ogretmen [1], Adam J. Bass[3,4], Kwok-Kin Wong[3,4], Anil K. Rustgi[5,6,7] & J. Alan Diehl [1]

The dysregulation of Fbxo4-cyclin D1 axis occurs at high frequency in esophageal squamous cell carcinoma (ESCC), where it promotes ESCC development and progression. However, defining a therapeutic vulnerability that results from this dysregulation has remained elusive. Here we demonstrate that Rb and mTORC1 contribute to Gln-addiction upon the dysregulation of the Fbxo4-cyclin D1 axis, which leads to the reprogramming of cellular metabolism. This reprogramming is characterized by reduced energy production and increased sensitivity of ESCC cells to combined treatment with CB-839 (glutaminase 1 inhibitor) plus metformin/phenformin. Of additional importance, this combined treatment has potent efficacy in ESCC cells with acquired resistance to CDK4/6 inhibitors in vitro and in xenograft tumors. Our findings reveal a molecular basis for cancer therapy through targeting glutaminolysis and mitochondrial respiration in ESCC with dysregulated Fbxo4-cyclin D1 axis as well as cancers resistant to CDK4/6 inhibitors.

[1] Department of Biochemistry and Molecular Biology, Hollings Cancer Center, Medical University of South Carolina, Charleston, SC 29425, USA. [2] Department of Drug Discovery and Biomedical Sciences, Medical University of South Carolina, Charleston, SC 29425, USA. [3] Department of Medical Oncology, Dana-Farber Cancer Institute, Boston, MA 02215, USA. [4] Department of Medicine, Brigham and Women's Hospital, Harvard Medical School, Boston, MA 02115, USA. [5] Department of Medicine, Division of Gastroenterology, University of Pennsylvania Perelman School of Medicine, Philadelphia, PA 19104, USA. [6] Abramson Cancer Center, University of Pennsylvania Perelman School of Medicine, Philadelphia, PA 19104, USA. [7] Department of Genetics, University of Pennsylvania Perelman School of Medicine, Philadelphia, PA 19104, USA. Correspondence and requests for materials should be addressed to J.A.D. (email: diehl@musc.edu)

Esophageal squamous cell carcinoma (ESCC) accounts for ~90% of esophageal cancer worldwide, and it remains an aggressive and lethal malignancy[1]. Current therapies have limited efficacy due to local invasion and lymphatic metastasis, which are common with late stage disease, highlighting the urgent need for second-line treatments[2]. Genome-wide screening has revealed numerous genetic alterations in ESCC, including inactivating mutations of *TP53*, Retinoblastoma protein (*Rb*) and *CDKN2A*, or activating mutations of *NFE2L2*, *NOTCH1/2*, *MLL2*, and *EP300*, or amplification of *cyclin D1*[3].

Cyclin D1-CDK4/6 promotes G$_1$ cell cycle progression through phosphorylation-dependent inactivation of Rb, the master gatekeeper of cell division. Rb phosphorylation promotes E2F activation and the transcription of target genes critical for cell cycle progression[4,5]. Cyclin D1-CDK4 also regulates cell migration, stress responses, and cellular metabolism[6–8]. Overexpression of cyclin D1, due to gene amplification and inactivation of its degradation machinery, (for example, loss of the Fbxo4 E3 ubiquitin ligase or mutations in the cyclin D1 degron), directly contributes to the development and progression of ESCC[9–11], supporting a model wherein targeting cyclin D1-CDK4/6 could be effective for ESCC patients. Recently, palbociclib (PD-0332991), the first FDA-approved small molecule CDK4/6 inhibitor, has undergone extensive investigations in pre-clinical studies and clinical trials in a series of human malignancies, including esophageal cancer[12–14]; however, intrinsic and acquired resistance remains a major concern, and one of the mechanisms has been linked to *Rb* loss[15,16].

Recent investigations of the *c-Myc* oncogene have highlighted the importance of Glutamine (Gln) metabolism in the survival and proliferation of tumor cells, which is defined as Gln-addiction or Gln-dependency[17,18]. Gln is metabolized by a process known as glutaminolysis, whereby it is converted to glutamate, and subsequently to α-ketoglutarate (α-KG) for energy production[19]. Oncogenes and tumor suppressors can control Gln metabolism through regulating the expression and/or activation of glutaminase (GLS), the key rate-limiting enzyme for glutaminolysis[17,20,21]. Two isoforms of GLS have been identified: GLS1 and GLS2. Knockdown or chemical suppression of GLS1 typically induces apoptosis, suppresses cell proliferation and tumor growth[20,22]. Besides *c-Myc* oncogene, Rb loss is also associated with cellular dependency on Gln[23], emphasizing the therapeutic potential by targeting these genetic predispositions. However, it remains unclear whether Rb loss-mediated Gln-addiction is subject to cyclin D1 regulation. Given that Rb is rarely lost in ESCC, while Fbxo4 loss or *cyclin D1* amplification occurs at a high frequency, it is important to fill this knowledge gap in order to develop therapies for ESCC that may also be efficient for other tumors with dysregulation of this signaling pathway.

This work demonstrates the contribution of Fbxo4 loss and hyperactivation of cyclin D1-CDK4/6 kinases to Gln-addiction in ESCC cells. We demonstrate that cyclin D1 overexpression, either as a consequence of direct mutation, or loss of its regulatory E3 ubiquitin ligase Fbxo4, results in Gln-addiction. The dysregulation of Fbxo4-cyclin D1 axis leads to mitochondrial dysfunction and Gln-addiction. Clinically, combined treatment with CB-839, a GLS1 inhibitor currently being evaluated in clinical trials, and metformin/phenformin effectively induces apoptosis and suppresses cell proliferation in vitro and in vivo; furthermore, combined treatment also shows promising therapeutic potential in tumors resistant to CDK4/6 inhibitors.

## Results

**Dysregulated Fbxo4-cyclin D1 drives Gln-addiction**. Gln-addiction has been associated with overexpression of c-Myc[17,18];

however, its role has not been evaluated in cells harboring Fbxo4 mutation or cyclin D1 overexpression, which frequently occurs in human ESCC[10,24]. To address this question, we set out to determine whether Fbxo4 impacts cellular dependency on Gln. *Fbxo4*+/+ or −/− mouse embryonic fibroblasts (MEFs) were exposed to medium without Gln. Gln-depletion triggered increased cleavage of both PARP and caspase-3, suggesting increased apoptosis of *Fbxo4*−/− versus +/+ MEFs (Fig. 1a, b). Elevated Gln uptake was apparent in *Fbxo4*−/− MEFs (Supplementary Fig. 1a), indicating Fbxo4 loss drives increased Gln uptake and enhances Gln-dependency. To further assess how Fbxo4 regulates cellular response to Gln availability, a Gln-depletion/re-supplementation approach was applied, wherein cells were subjected to Gln-depletion for 24 h, followed by Gln re-addition. Analogous with the previous observation, increased apoptosis was observed in *Fbxo4*−/− MEFs (Supplementary Fig. 1b). To determine whether the sensitivity to Gln-depletion reflected Fbxo4 E3 ligase function, WT Fbxo4 or Fbxo4 ΔN, ΔF, ΔC2, and ΔC3 were transduced into *Fbxo4*−/− MEFs[10,25]. WT Fbxo4, but not nonfunctional mutants, reduced cellular dependency on Gln (Fig. 1c, d), suggesting substrates of Fbxo4 may drive Gln-addiction.

As c-Myc promotes Gln-addiction[17,18], we assessed c-Myc levels in *Fbxo4*+/+ and −/− MEFs. *Fbxo4*−/− MEFs have lower c-Myc protein levels upon Gln-withdrawal (Supplementary Fig. 1c), demonstrating that c-Myc expression does not contribute to Gln-addiction in this genetic background. Previous work revealed the role of Rb loss in Gln-addiction[23]; importantly, Rb can be inactivated through phosphorylation mediated by the cyclin D1-CDK4/6 kinase; moreover, cyclin D1 is a well established Fbxo4 substrate[26]. Therefore, we determined the importance of cyclin D1 in Gln-addiction observed in *Fbxo4*−/− cells. We generated MEFs from *Fbxo4* and *cyclin D1* double knockout mice (Supplementary Fig. 1d). *Fbxo4* and *cyclin D1* double knockout MEFs exhibited lower apoptosis triggered by Gln-depletion relative to *Fbxo4* single knockout MEFs (Fig. 1e). In addition, ectopic expression of WT cyclin D1, or a stabilized Fbxo4-resistant cyclin D1 mutant, D1T286A, greatly sensitized cells to Gln restriction (Fig. 1f and Supplementary Fig. 2a, b). The above findings indicate cyclin D1 is required and sufficient for Gln-addiction in cells with inactive *Fbox4*. Furthermore, Gln uptake was also increased by ectopic cyclin D1 (Supplementary Fig. 2c). Palbociclib (PD-0332991), a highly specific CDK4/6 kinase inhibitor, suppressed cell apoptosis in cells with cyclin D1 overexpression (Fig. 1g and Supplementary Fig. 2b), supporting our hypothesis that cyclin D1-CDK4/6 drives Gln-addiction.

**Fbxo4-cyclin D1 drives Gln-addiction in ESCC cells**. Loss of function mutation of *Fbxo4* leads to cyclin D1 accumulation, contributing to the development of human ESCC[10]; moreover, Fbxo4 loss results in susceptibility to upper gastrointestinal tumors in transgenic mice[27]. Gene set enrichment analysis (GSEA) highlighted the activation of cell cycle regulators and dysregulation of Gln metabolism genes in two independent studies when comparing ESCC with the normal esophageal tissues[28] (Fig. 2a, b and Supplementary Fig. 3a, b and Supplementary Tables 1–4). Additional analysis revealed the reprogramming of Gln metabolism genes in ESCC tissues (Supplementary Fig. 4). Oncomine analysis also highlighted the elevation of *GLS1* mRNA in human ESCC relative to normal tissues (Supplementary Fig. 3c–e), suggesting increased Gln metabolism in ESCC tumor tissue. Furthermore, SurvExpress survival analysis indicated the dysregulation of Gln metabolism genes correlates with poor prognosis of human esophageal cancer as well as Head and Neck SCC (Fig. 2c and Supplementary Fig. 5).

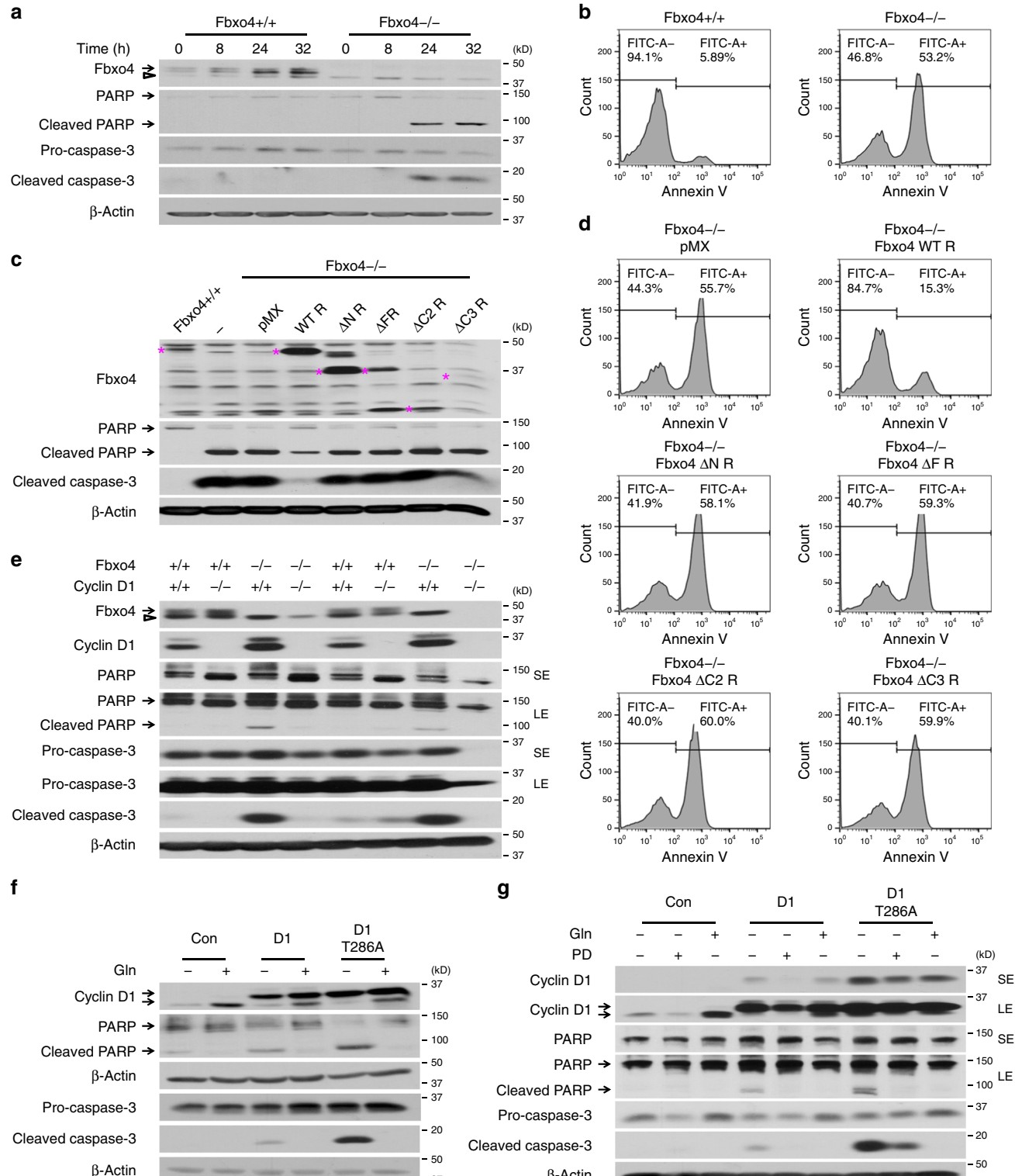

**Fig. 1** Dysregulated Fbxo4-cyclin D1 axis leads to Gln-addiction. **a** Western analysis for cleaved PARP and cleaved caspase-3 in *Fbxo4−/−* mouse embryonic fibroblasts (MEFs) following Gln-depletion. **b** FACS for Annexin V-positive cells following 48 h Gln-depletion. **c** Western analysis for cleaved PARP and cleaved caspase-3 following 24 h Gln-depletion in *Fbxo4−/−* MEFs with ectopic expression of Fbxo4 WT, ΔN, ΔF, ΔC2, or ΔC3. Magenta stars indicate Fbxo4 bands. **d** FACS for Annexin V-positive cells following 48 h Gln-depletion. **e** *Cyclin D1* knockout antagonizes apoptosis in a *Fbxo4−/−* background following 24 h Gln-depletion. In order to show cyclin D1 expression, cyclin D1 blot was performed in medium with Gln, because Gln-depletion reduces endogenous cyclin D1 expression. **f** Overexpression of cyclin D1 promotes apoptosis in NIH3T3 cells upon 24 h Gln-depletion. **g** One micromolar PD-0332991 (PD) suppresses apoptosis induced by 24 h Gln-depletion in NIH3T3 cells with ectopic cyclin D1 or D1T286A. SE: short exposure; LE: long exposure. Arrow: specific band; open triangle: non-specific band

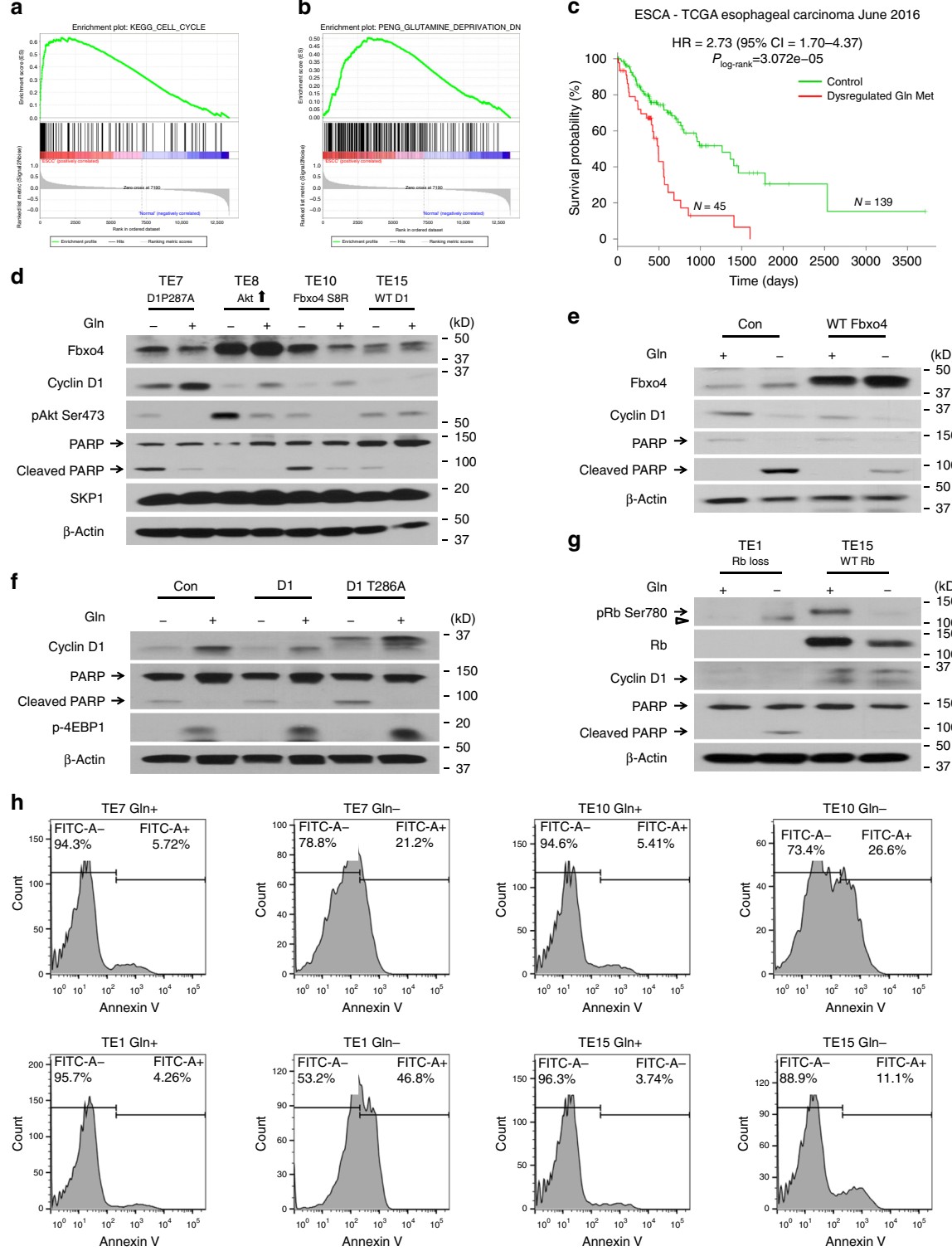

**Fig. 2** Dysregulated Fbxo4-cyclin D1 axis promotes Gln-addiction in esophageal squamous cell carcinoma (ESCC) cells. **a** Gene set enrichment analysis (GSEA) enrichment of ESCC versus normal tissues for gene set "Cell Cycle Regulation" in NCBI GEO dataset (GSE100942): NES = 1.4644697, FDR $q$-val = 0.45459393. **b** GSEA enrichment of ESCC versus normal tissues for gene set related to "Gln Metabolism" in NCBI GEO dataset (GSE100942): NES = 1.5255251, FDR $q$-val = 0.06451613. **c** Expression of Gln metabolism genes correlates with poor prognosis of ESCC patients. Dysregulated Gln Met: dysregulated Gln metabolism; HR: hazard ratio; CI: confidence interval. **d** Increased PARP cleavage following 24 h Gln-depletion in TE7 and TE10 versus TE15 cells. **e** WT Fbxo4 suppresses PARP cleavage following 24 h Gln-depletion in TE10 cells. **f** Overexpression of cyclin D1 or D1T286A promotes the sensitivity of TE15 cells to 24 h Gln-depletion. **g** Increased PARP cleavage in TE1 versus TE15 cells upon 24 h Gln-depletion. **h** FACS for Annexin V-positive cells with or without 48 h Gln-depletion in TE7, TE10, TE1, and TE15 cells. Arrow: band of interest; open triangle: non-specific band

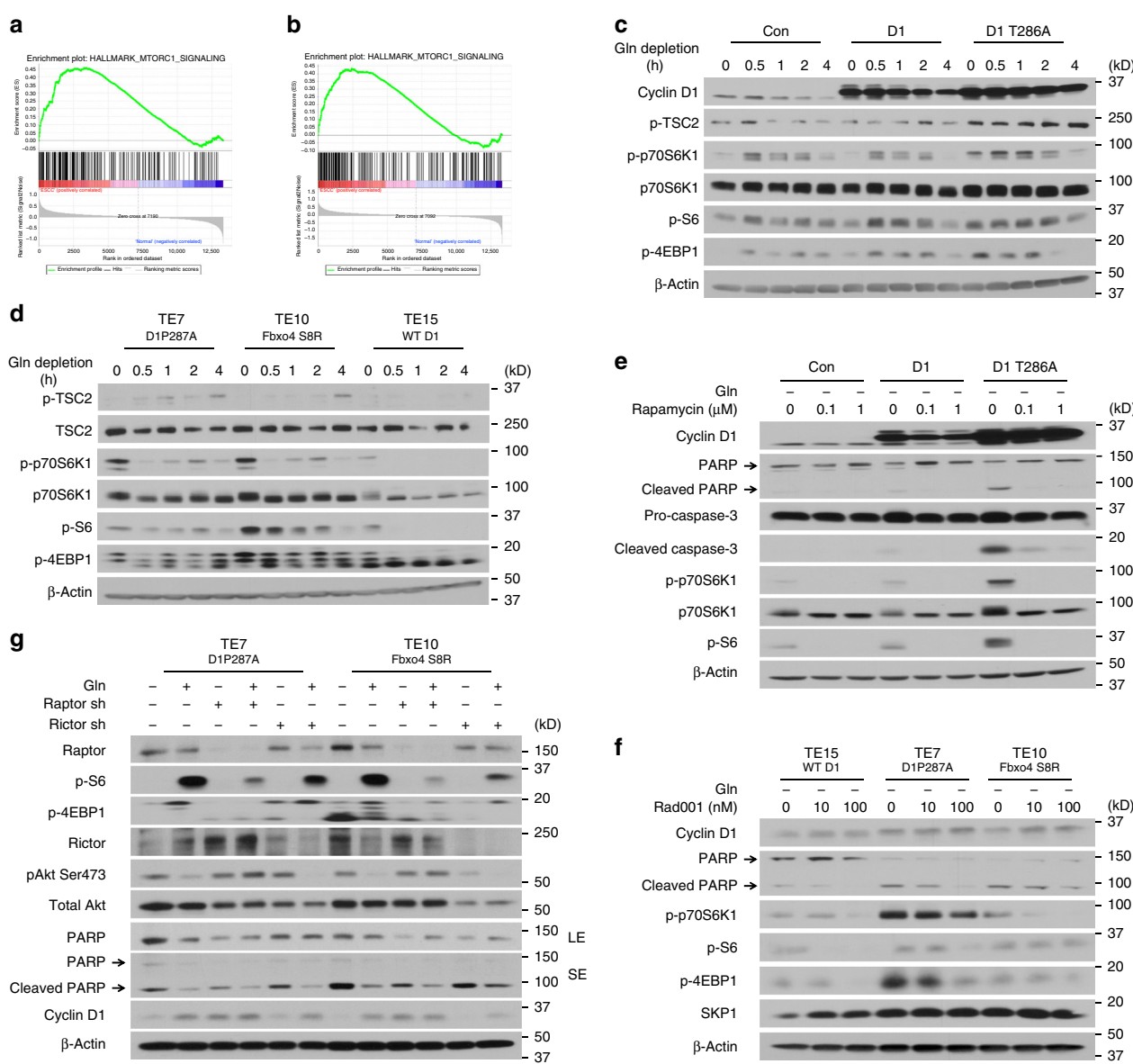

**Fig. 3** mTORC1 contributes to Gln-addiction in cells with dysregulated cyclin D1. **a, b** Gene set enrichment analysis (GSEA) enrichment plots for expression profiles of esophageal squamous cell carcinoma (ESCC) versus normal tissues for gene sets correlated to mTORC1 signaling in NCBI GEO datasets: **a** GSE100942—NES = 1.4599689, FDR q-val = 0.12097056 and **b** GSE20347—NES = 1.269795, FDR q-val = 0.3669077. **c** Western blot for the indicated mTORC1 effectors in NIH3T3 cells expressing cyclin D1 or D1T286A. **d** Western blot analysis for mTORC1 effectors in TE7, TE10, and TE15 cells. **e** Rapamycin compromises mTORC1 activation and apoptosis induced by ectopic cyclin D1 expression in NIH3T3 cells upon 24 h Gln-depletion. **f** Rad001 inhibits mTORC1 activation and apoptosis in TE7 and TE10 cells upon 24 h Gln-depletion. **g** Raptor knockdown suppresses apoptosis in TE7 and TE10 cells upon 24 h Gln-depletion. SE: short exposure; LE: long exposure. Arrow: specific band

To corroborate the above genome-wide findings, five representative ESCC cells were chosen to investigate Gln-dependency: TE7 (WT Fbxo4, cyclin D1P287A), TE8 (WT Fbxo4, WT cyclin D1, Akt hyperactivation), TE10 (Fbxo4 S8R, WT cyclin D1), TE15 (WT Fbxo4, WT cyclin D1), and TE1 (Rb deficient). Gln restriction triggered more PARP cleavage and apoptosis in TE7 and TE10 than in TE15 cells (Fig. 2d, h), consistent with the notion that dysregulation of Fbxo4-cyclin D1 promotes Gln-dependency. Importantly, *c-Myc* knockdown did not reduce Gln-addiction in TE7 or TE10 cells (Supplementary Fig. 6), suggesting cyclin D1 rather than c-Myc drives Gln-addiction in ESCC cells. Consistently, TE7 and TE10 cells exhibit increased Gln uptake relative to TE15 cells (Supplementary Fig. 7a). Furthermore, WT Fbxo4 not only reduced cyclin D1 levels but also compromised apoptosis triggered by Gln restriction in TE10 cells (Fig. 2e and

Supplementary Fig. 7b). To test whether cyclin D1 is sufficient to drive cellular Gln-dependency, cyclin D1 or D1T286A (a stable oncogenic D1 mutant analogous to D1P287A), were ectopically expressed in TE15 cells. D1T286A expression drove apoptosis of TE15 cells cultured in Gln-free medium (Fig. 2f and Supplementary Fig. 7c).

TE1 cells are Rb deficient, resulting in the bypass of cell cycle control that depends on cyclin D1-CDK4/6 complex[29,30]; Rb loss compromised cyclin D1 expression, but drove the sensitivity to Gln restriction (Fig. 2g, h). Likewise, *Rb* knockdown promoted Gln-addiction in TE15 cells that mimics the findings in TE1 cells (Supplementary Fig. 7d, e). Taken together, these data reinforce the role of Fbxo4-cyclin D1 axis in driving Gln-addition in ESCC cells, and Rb is an important contributing factor.

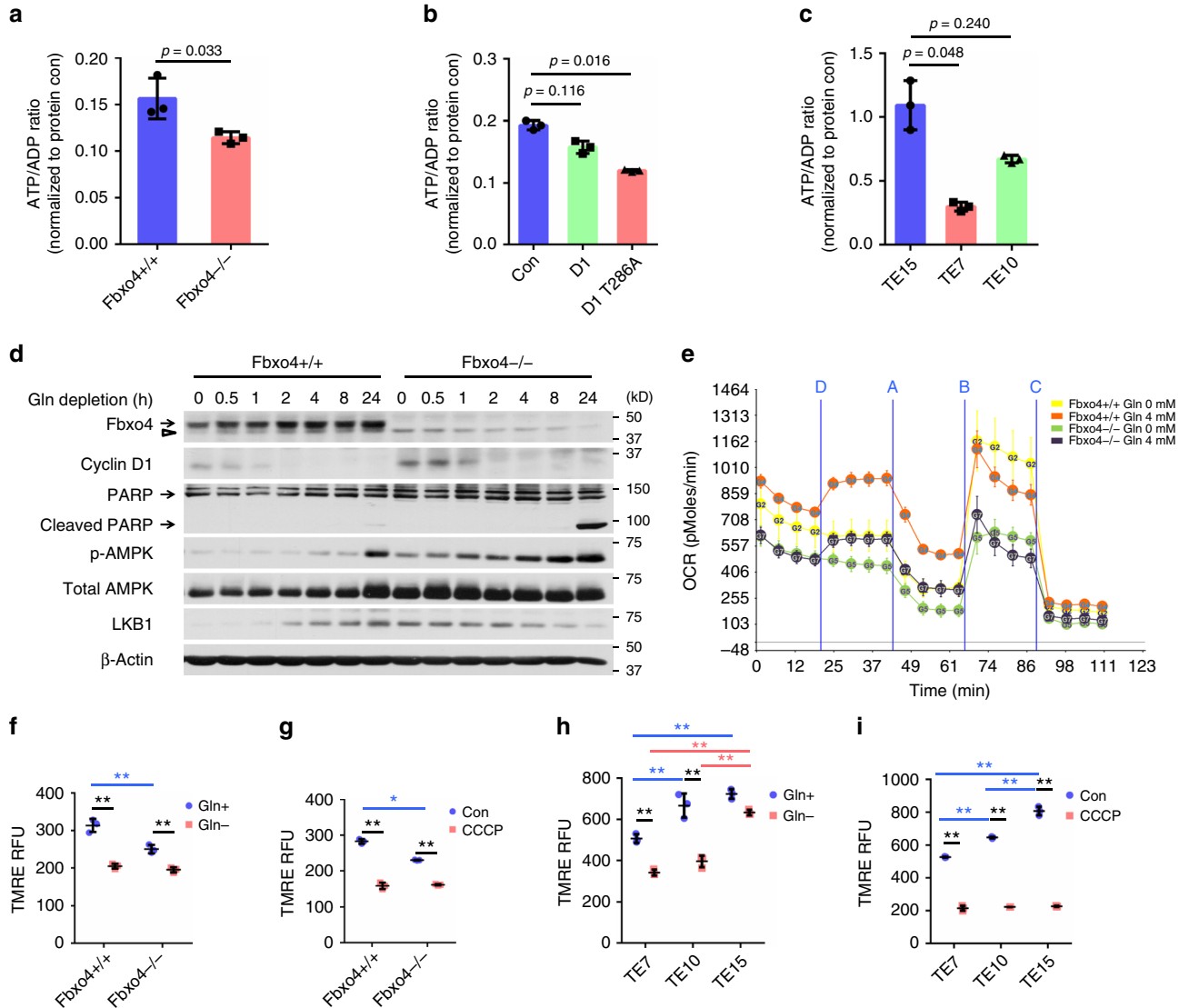

**Fig. 4** Cyclin D1 accumulation promotes mitochondrial dysfunction. **a–c** ATP/ADP ratio in *Fbxo4+/+* and −/− mouse embryonic fibroblasts (MEFs) (**a**), NIH3T3 cells (**b**), and esophageal squamous cell carcinoma (ESCC) cells (**c**) detected by ApoSENSOR™ ADP/ATP ratio bioluminescent assay. **a** Data represent as mean ± s.d., two-tailed Student *t*-test was used to compare means (n = 3); **b**, **c** Data represent as mean ± s.d., one-way ANOVA was used to compare means with Bonferroni as Post Hoc test (n = 3); *p*-values are listed. **d** Western analysis for the indicated proteins in *Fbxo4+/+* versus −/− MEFs upon Gln-depletion. Arrow: band of interest; open triangle: non-specific band. **e** Seahorse analysis for *Fbxo4+/+* and −/− MEFs; data were normalized to protein concentration. Injections: Port D: Gln; Port A: Oligomycin; Port B: Carbonyl cyanide-4-(trifluoromethoxy)phenylhydrazone (FCCP) and Port C: Antimycin A & Rotenone. **f, g** Mitochondrial membrane potential analysis of *Fbxo4+/+* and −/− MEFs; **g** Carbonyl cyanide m-chlorophenyl hydrazone (CCCP) treatment as positive control. **h, i** Mitochondrial membrane potential analysis of TE7, TE10, and TE15 cells; **i** CCCP treatment as positive control. TMRE, tetramethylrhodamine, ethyl ester; RFU, relative fluorescence units. All data in **f–i** represent as mean ± s.d., one-way ANOVA was used to compare means with Bonferroni as Post Hoc test (n = 3). *$p < 0.05$; **$p < 0.01$

**mTORC1 activation also contributes to Gln-addiction.** Mammalian target of rapamycin complex 1 (mTORC1), a critical sensor of extracellular stimuli and nutrient levels, functions as a central regulator to balance cell proliferation/growth and the availability of nutrients[31,32]. Dysregulation of mTORC1 signaling has been observed in pathological conditions, including cancer, obesity, diabetes, and neuro-degeneration[33]. Cyclin D1-CDK4/6 phosphorylates and inactivates TSC1/2, resulting in the activation of mTORC1 signaling[34,35]. Consistent with frequent overexpression of cyclin D1 in ESCC, GSEA revealed mTORC1 activation in ESCC relative to normal tissues (Fig. 3a, b and Supplementary Tables 5–6). To test whether mTORC1 is induced by cyclin D1, NIH3T3 or ESCC cells were exposed to Gln-withdrawal. Analysis revealed that cyclin D1 overexpression

triggered increased phosphorylation of p70S6K1, S6 and 4EBP1 in NIH3T3 cells (Fig. 3c). In addition, dysregulation of Fbxo4-cyclin D1 also promoted more mTORC1 activation in TE7 and TE10 cells compared to TE15 cells; furthermore, increased basal mTORC1 activation was also detected in TE7 and TE10 cells (Fig. 3d). These data demonstrate mTORC1 signaling is hyperactivated in cells with cyclin D1 overexpression.

We next assessed the relevance of elevated mTORC1 activation with Gln-addiction in cells harboring dysregulated Fbxo4-cyclin D1 axis. Cells were exposed to Gln-free medium; meanwhile, compounds or genetic manipulations were applied to compromise mTORC1 activity. Consistent with mTORC1 inhibition, both rapamycin and Rad001, partially suppressed cell apoptosis as determined by reduced cleavage of both PARP and caspases-3

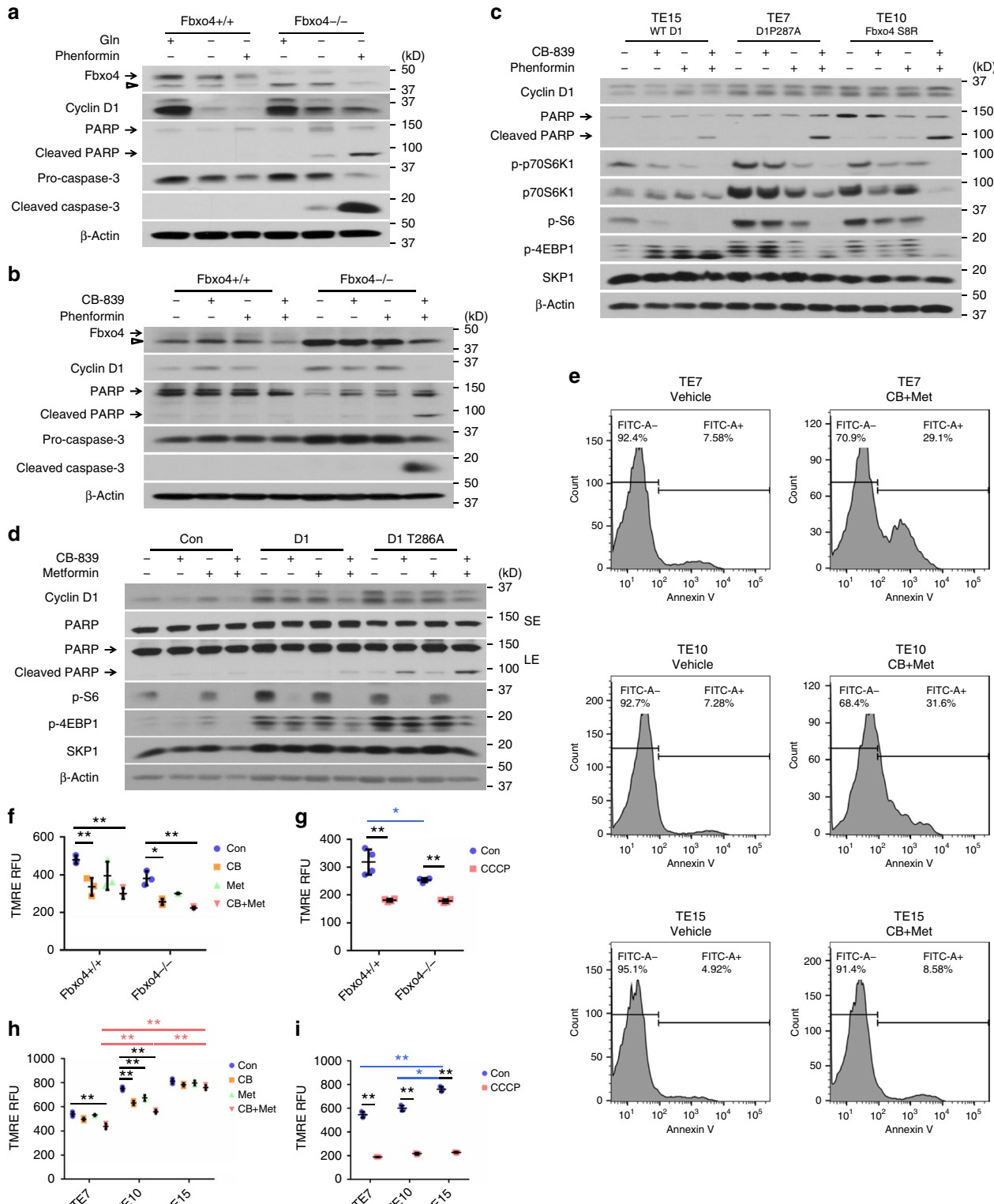

**Fig. 5** CB-839 plus Phenformin/Metformin synergistically induces apoptosis. **a** Western analysis of PARP and caspase-3 cleavage in *Fbxo4−/−* versus +/+ mouse embryonic fibroblasts (MEFs) treated with Phenformin following 24 h Gln-depletion. **b** Western analysis for the indicated proteins in *Fbxo4+/+* versus −/− MEFs treated with CB-839 plus phenformin for 24 h. **c** Western analysis for PARP cleavage following 24 h treatment with CB-839 and/or phenformin in TE7, TE10, and TE15 cells. **d** Combined treatment for 24 h induces more apoptosis in TE15 cells with ectopic cyclin D1 or D1T286A expression. SE: short exposure; LE: long exposure. **e** FACS for Annexin V-positive cells following combined CB-839 (CB) and metformin (Met) treatment for 48 h. **f**, **g** Mitochondrial membrane potential analysis of *Fbxo4+/+* and −/− MEFs with the indicated treatments; **g** CCCP treatment as positive control. **h**, **i** Mitochondrial membrane potential analysis of TE7, TE10, and TE15 cells with indicated treatments; **i** CCCP treatment as positive control. All data represent as mean ± s.d., one-way ANOVA was used to compare means with Bonferroni as Post Hoc test ($n = 3$). $*p < 0.05$; $**p < 0.01$. CB-839 concentration is 10 μM and phenformin/metformin concentration is 1 mM for panels **a**–**i**. Arrow: specific band; open triangle: non-specific band

(Fig. 3e, f and Supplementary Fig. 8a, b). In addition, *Raptor* knockdown also compromised cell apoptosis (Fig. 3g and Supplementary Figs. 8c and 9), emphasizing mTORC1 partially drives Gln-addiction in cells with dysregulated Fbxo4-cyclin D1 axis.

**Cyclin D1 drives mitochondrial dysfunction.** Balanced energy production in accordance with proliferation is critical for cellular homeostasis under nutrient-depleted conditions, which frequently occur during tumor progression. As Gln serves as an energy metabolite[36], we assessed the energetic status of cells with dysregulated Fbxo4-cyclin D1 axis in complete medium supplemented with Gln. *Fbxo4−/−* MEFs exhibited decreased ATP/ADP ratio (Fig. 4a). Likewise, overexpression of cyclin D1 or D1T286A, either of which increases CDK4/6 catalytic activity[37], reduced the ATP/ADP ratio with oncogenic D1T286A having a more profound effect (Fig. 4b). In addition, TE10 cells with mutant Fbxo4 exhibited a decreased ATP/ADP ratio, as did TE7 cells harboring D1P287A, a mutant refractory to Fbxo4-mediated degradation (Fig. 4c). Moreover, activation of the AMPK pathway was also observed upon Gln-depletion (Fig. 4d), supporting reduced ATP/ADP ratio in *Fbxo4−/−* MEFs. As a downstream factor of Gln and an alternative energy source, dimethyl 2-oxoglutarate (DM-α-KG), partially rescued cell apoptosis upon Gln-withdrawal; however, asparagine (a nitrogen source) and scavengers of reactive oxygen species (ROS) had no effects (Supplementary Fig. 10a–d), further highlighting the importance of Gln as an energy source to keep cell survival.

As energy production from Gln is coupled with oxidative phosphorylation (OXPHOS), Seahorse analysis was used to measure the cellular oxygen consumption rate (OCR). *Fbxo4 −/−* MEFs exhibited reduced oxygen consumption relative to +/+ counterparts (Fig. 4e), suggesting Fbxo4 loss compromises energy production through suppressing OXPHOS. To further investigate mitochondrial function, the membrane potential of mitochondria was assessed. *Fbxo4−/−* MEFs exhibited a lower mitochondrial potential than +/+ counterparts (Fig. 4f, g). Similar results were observed when comparing TE7 and TE10 cells with TE15 cells (Fig. 4h, i), demonstrating this observation is not restricted to fibroblasts. Moreover, ectopic cyclin D1 also compromised mitochondrial membrane potential in NIH3T3 cells (Supplementary Fig. 10e, f).

**Therapeutically targeting dysregulated Fbxo4-cyclin D1.** Gln-addiction and defective mitochondrial respiration suggests the therapeutic potential of targeting both glutaminolysis and mitochondrial respiration in tumors with dysregulated Fbxo4-cyclin D1 axis. We, therefore, targeted OXPHOS with either phenformin or metformin, both of which belong to the biguanide family and can inhibit mitochondrial respiration; importantly, metformin is clinically used to treat type II diabetes[38]. Phenformin induced more apoptosis in *Fbxo4−/−* MEFs challenged by Gln-depletion or GLS1 inhibition (Fig. 5a, b). Therefore, a combined treatment was proposed by applying GLS1 inhibitor, CB-839, and metformin/phenformin together to test the therapeutic potential. Consistently, combined treatment induced more cell apoptosis in NIH3T3 and TE15 cells with ectopic cyclin D1 expression and in TE7 or TE10 cells harboring dysregulated Fbxo4-cyclin D1 (Fig. 5c–e and Supplementary Figs. 11a, b and 12) than single agent alone. Apoptosis induced by combined treatment was antagonized by re-expression of WT Fbxo4 (Supplementary Fig. 11c, d). In addition, combined treatment also significantly compromised the mitochondrial potential in cells with cyclin D1 accumulation (Fig. 5f–i). Taken together, these data provide a

**Table 1 The EC50s of listed compounds with either single or combined treatment**

| Cell | Single[a] CB-839 (μM) | Single[a] metformin (mM) | Combined[b] CB-839 (μM) | Combined[b] metformin (mM) |
|---|---|---|---|---|
| TE7 (D1P287A)[c] | 4.75 | 7.29 | 0.75 | 1.20 |
| TE10 (Fbxo4 S8R)[c] | 5.05 | 8.37 | 0.92 | 1.47 |
| TE15 (WT D1)[c] | 16.52 | 9.88 | 1.77 | 2.83 |
| TE7PDR | 0.23 | 3.64 | 0.09 | 0.144 |
| TE10PDR | 0.21 | 3.78 | 0.09 | 0.144 |

[a]Single, single treatment with CB-839 or metformin
[b]Combined, combined treat with both CB-839 and metformin
[c]For TE7, TE10, and TE15 cells, the information in the parentheses indicates gene status

molecular basis for treating tumors with dysregulated Fbxo4-cyclin D1.

**Synergistic effects of combined treatment.** Given the effects of the combined treatment on cultured cells, the half maximal effective concentrations (EC50s) for CB-839 and/or metformin were assessed in TE7, TE10, and TE15 cells. For single treatment, the EC50 for either CB-839 or metformin was uniformly lower in TE7 and TE10 cells relative to TE15 cells (Table 1 and Supplementary Fig. 13). For combined treatment, the EC50s for TE7 and TE10 cells were lower than that for TE15 cells (Table 1 and Supplementary Fig. 13). To discriminate between synergistic versus additive effects, the combination indexes were calculated (Supplementary Fig. 13j–l). The combined treatment exhibited synergism at low concentration, with a stronger impact on TE7 and TE10 cells; the EC50s of CB-839 were 0.75 and 0.92 μM, respectively, compared to 1.77 μM for TE15 cells. To define the lowest concentration with effective suppression, different concentration combinations were applied to ESCC or NIH3T3 cells. The analyses demonstrated that cells with elevated cyclin D1 were more sensitive to combined treatment, 5 μM CB-839 and 0.5 mM metformin exhibited better suppressing effects in TE7 and TE10 than that in TE15 cells, while similar effects were also revealed in NIH3T3 cells (Supplementary Fig. 14a–f). In addition, combined treatment also effectively suppressed colony formation of NIH3T3 cells with ectopic cyclin D1 expression (Supplementary Fig. 14g).

**Combined treatment reduces tumor burden in vivo.** To assess the efficacy of combined treatment in vivo, two cell lines: TE7 (mutant cyclin D1P287A) and TE15 (WT cyclin D1) were chosen. Cells were subcutaneously injected into flank regions of athymic nude, nu/nu mice. Once tumors were established, mice were treated with vehicle, CB-839, metformin or their combination. CB-839 was orally gavaged at 200 mg/kg twice per day, and metformin was administrated i.p. (intraperitoneally) at 250 mg/kg once per day. Mice tolerated the treatment with only a minor variation in body weight and glucose levels (Supplementary Fig. 15a–d). CB-839 plus metformin exhibited stronger suppressive effects on growth of TE7 relative to TE15 tumor xenografts (Fig. 6a, b and Supplementary Fig. 15e, f), consistent with analysis of cultured cells. Morphological analysis revealed that both TE7 and TE10 cells formed squamous cell carcinoma (Fig. 6c and Supplementary Fig. 15g). Nuclear magnetic resonance spectroscopy (NMR) demonstrated that CB-839 triggered Gln accumulation in liver tissues and tumor xenografts (Supplementary Fig. 16), supporting the suppression of glutaminolysis and on-target activity. In addition, pathological analyses revealed stronger impacts on cell proliferation (Ki-67 index) and apoptosis (cleaved

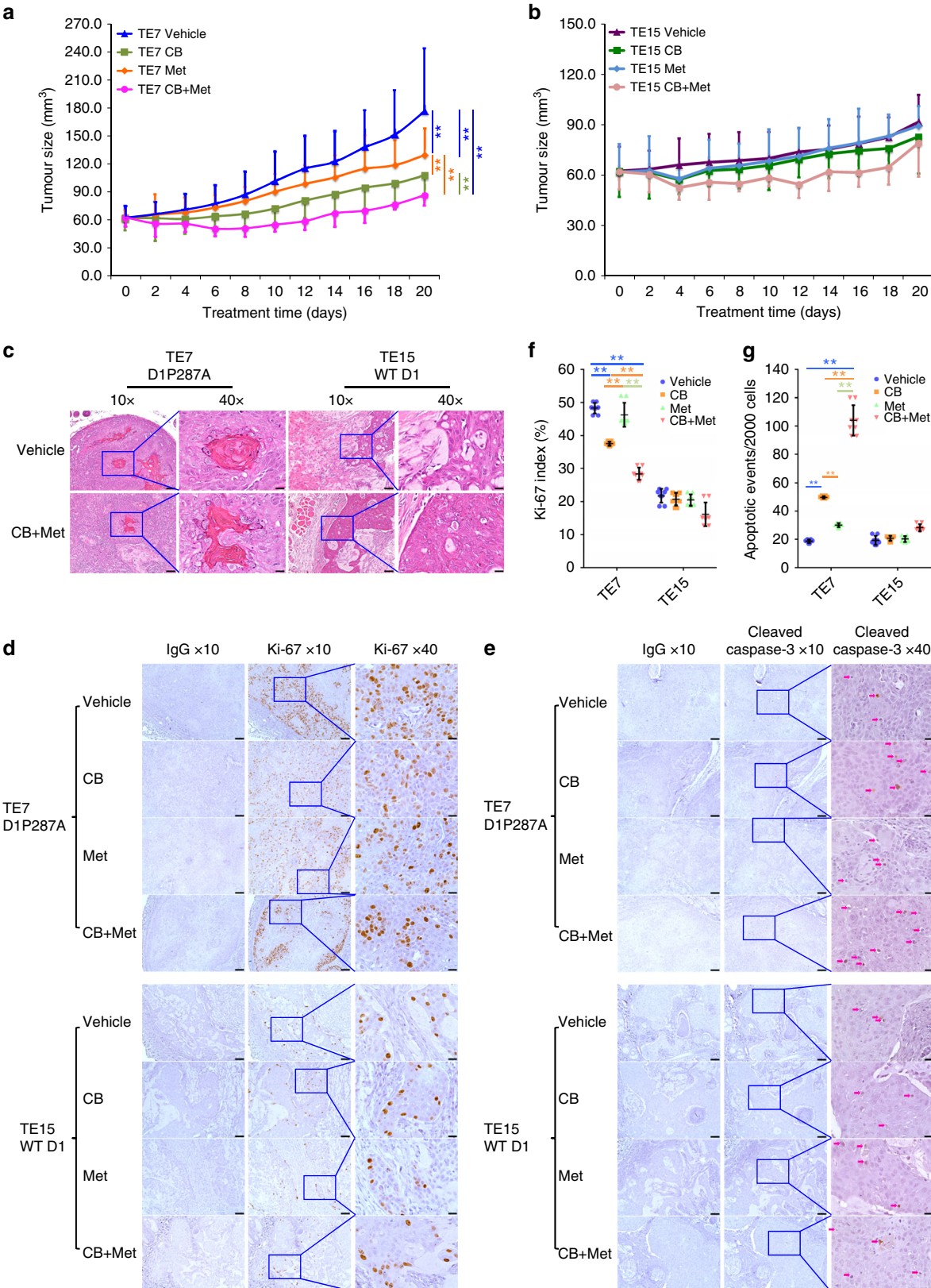

**Fig. 6** Combined treatment inhibits xenograft growth and induces apoptosis in vivo. **a**, **b** Growth curve of TE7 (**a**) and TE15 (**b**) xenografts. Data represent as mean ± s.d., two-way ANOVA was used to compare means with Bonferroni as Post Hoc test ($n = 8$). **$p < 0.01$. **c** H&E staining sections from TE7 and TE15 xenografts. **d** IHC staining of Ki-67 in xenograft tissues. **e** IHC staining of cleaved caspase-3 in xenograft tissues, magenta arrow indicates positive cells. **f** Comparison of Ki-67 index among different groups. **g** Quantification of cleaved caspase-3 positive cells per 2000 cells. All data in **f** and **g** represent as mean ± s.d., one-way ANOVA was used to compare means with Bonferroni as Post Hoc test ($n = 8$). **$p < 0.01$. Scale bar, 10 µm

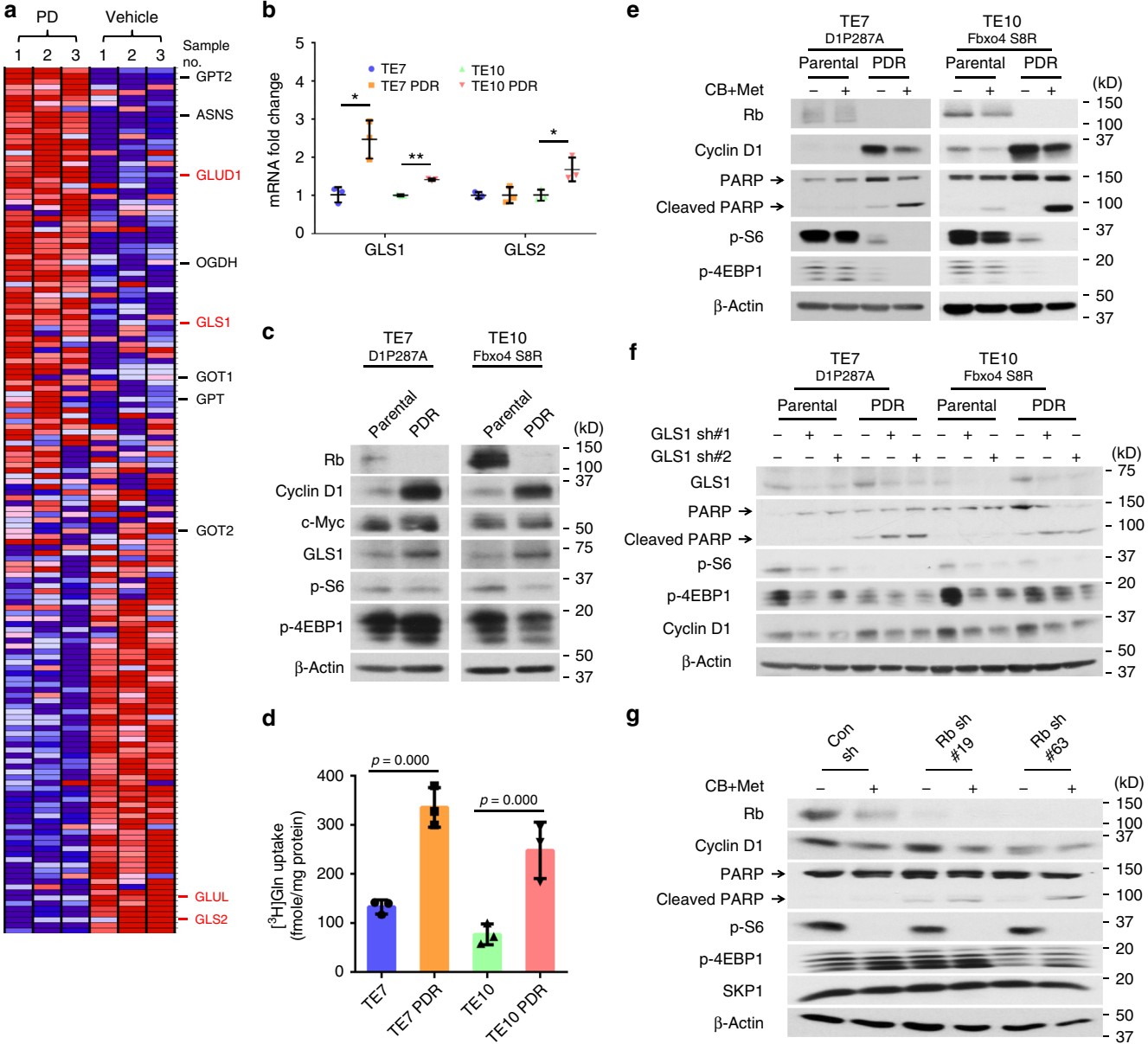

**Fig. 7** Combined treatment overcomes CDK4/6 inhibitor resistance in esophageal squamous cell carcinoma (ESCC) cells. **a** Blue-Pink O' Gram in the Space of the Analyzed Gene Set with NCBI GEO (GSE40513), the comparison of Gln metabolism genes between palbociclib and vehicle-treated mouse breast cancer V720 cells. Glutamate-ammonia ligase (GLUL), glutamate dehydrogenase (GLUD), asparagine synthetase (ASNS), glutamic-oxaloacetic transaminase (GOT), and glutamic-pyruvic transaminase (GPT). Red color indicates gene upregulation; blue color indicates gene downregulation. **b** The expression of *GLS1* and *GLS2* mRNAs in parental and PDR cells. Data represent as mean ± s.d., two-tailed Student *t*-test was used to compare means ($n =$ 3). *$p < 0.05$, **$p < 0.01$. **c** Western blot analysis of GLS1 levels in parental and PDR cells. **d** PDR ESCC cells exhibit increased [$^3$H]Gln uptake relative to parental counterparts ($1 \times 10^5$ cells used). Data represent as mean ± s.d., one-way ANOVA was used to compare means with Bonferroni as Post Hoc test ($n = 3$); *p*-values are listed. **e** CB-839 plus metformin treatment for 24 h induces more apoptosis in both TE7PDR and TE10PDR cells. **f** *GLS1* knockdown increases apoptosis in both TE7PDR and TE10PDR cells. **g** *Rb* knockdown induces cell apoptosis in TE15 cells upon combined treatment for 24 h. Arrow: band of interest

caspase-3 staining) in TE7 xenografts than that in TE15 tumors (Fig. 6d–g). Taken together, these data support the combined treatment can effectively reduce tumor burden through interfering with cell proliferation and survival.

**Combined treatment overcomes Palbociclib resistance.** Rb loss induces resistance to CDK4/6 inhibitors, such as, palbociclib or ribociclib[23,39]. Moreover, investigation of genome-wide datasets revealed the reprogramming of Gln metabolism genes upon

either palbociclib treatment or *CDK4/6* knockdown[40,41] (Fig. 7a and Supplementary Fig. 17a) using a gene set related to "Gln Metabolism" (Supplementary Table 7). To address whether palbociclib-resistant (PDR) cells exhibit characteristics of Gln-addiction, TE7 and TE10 cells were exposed to medium with 1 μM palbociclib for extended periods until PDR lines of TE7 and TE10 cells were established[42]; flow cytometry revealed these cells successfully overcame cell cycle arrest-induced by palbociclib (Supplementary Fig. 18). Phenotypically, both TE7PDR and TE10PDR cells demonstrated loss of Rb, and upregulation of

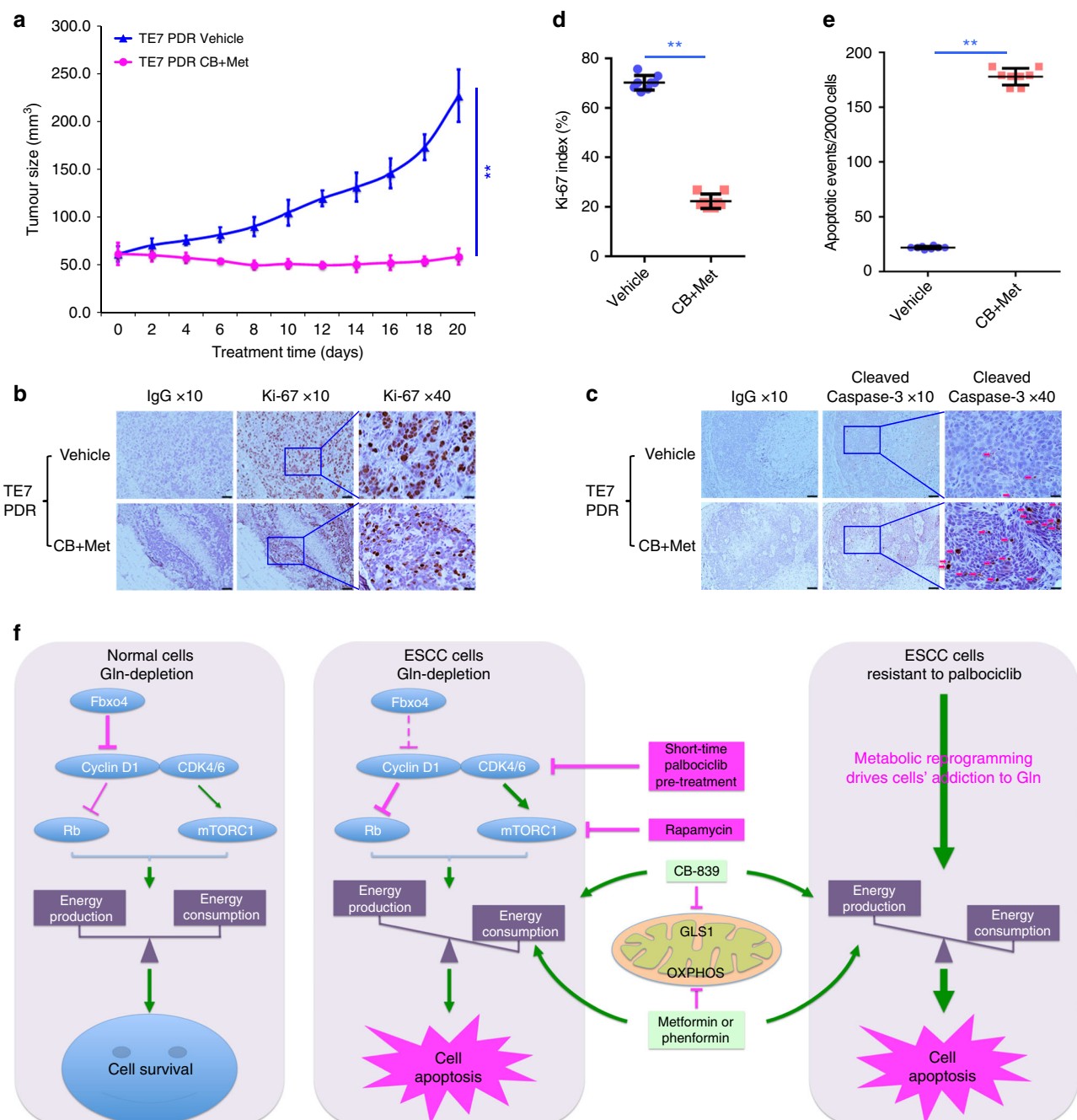

**Fig. 8** Combined treatment suppresses TE7PDR xenograft growth and induces cell apoptosis in vivo. **a** Growth curve of TE7PDR xenografts in nude mice. Data represent as mean ± s.d., two-way ANOVA was used to compare means with Bonferroni as Post Hoc test ($n = 8$). **$p < 0.01$. **b** IHC staining of Ki-67 in xenograft tissues. **c** IHC staining of cleaved caspase-3 in xenograft tissues, magenta arrow indicates positive cells. **d** Comparison of Ki-67 index between vehicle and treated groups. **e** Quantification of cleaved caspase-3 positive cells per 2000 cells between vehicle and treated groups. All data in **d** and **e** represent as mean ± s.d., two-tailed Student $t$-test was used to compare means ($n = 8$). **$p < 0.01$. Scale bar, 10 μm. **f** Schematic illustration of the working model. In normal cells, the homeostatic regulation of Fbxo4-cyclin D1 axis keeps the downstream pathway balanced; however, in tumor cells, dysregulated Fbxo4-cyclin D1 axis suppresses Rb and hyperactivates mTORC1, leading to the imbalance between energetic production and consumption, and finally, Gln-addiction. By targeting this genetic vulnerability, combined CB-839 and metformin/phenformin disrupts the metabolic balance, leading to cell apoptosis and the suppression of cell proliferation. In addition, palbociclib-resistant ESCC cells demonstrate metabolic reprogramming characterized by Gln-addiction, resulting in increased sensitivity to combined treatment. This model provides promising therapeutic targets for cancer treatment

GLS1 that was independent of c-Myc expression (Fig. 7b, c and Supplementary Fig. 17b). PDR cells also exhibited increased Gln uptake relative to parental counterparts (Fig. 7d). In accordance with these characteristics, both TE7PDR and TE10PDR cells were more sensitive to Gln-depletion (Supplementary Fig. 17c, d), combined treatment (Fig. 7e and Supplementary Fig. 19a), or

*GLS1* knockdown (Fig. 7f), indicating the feasibility of killing these cells using CB-839 plus metformin. As palbociclib-resistant TE15 cells could not be established through extended culture, *Rb* knockdown was utilized to mimic palbociclib resistance. Consistently, combined treatment also induced more cell apoptosis in cells with *Rb* knockdown (Fig. 7g and Supplementary Fig. 19b). In

addition, both TE7PDR and TE10PDR cells showed lower EC50s comparing to their parental counterparts (Table 1 and Supplementary Fig. 20), indicating combined treatment can effectively overcome palbociclib resistance.

To corroborate the above findings in cell culture, TE7PDR cells were injected into nude mice in order to test the efficacy of combined treatment in PDR tumors in vivo. Mice tolerated the treatment with only minor alteration of body weight and glucose levels (Supplementary Fig. 21a, b). TE7PDR cells formed squamous cell carcinoma xenografts that responded better to combined treatment than their parental counterparts (Figs. 6a and 8a, and Supplementary Fig. 21c, d). Consistent with the growth pattern, reduced cell proliferation and increased cell apoptosis were revealed (Fig. 8b–e), indicating CB-839 plus metformin effectively suppresses the growth of PDR xenografts.

## Discussion

Reprogramming of cellular metabolism is regarded as a hallmark of tumorigenesis[43,44]. Two major forms of dysregulated metabolism are typically observed: the Warburg effect and active glutaminolysis[45]. Glutaminolysis or Gln-addiction is driven by oncogenes like c-Myc, N-Myc, or receptor tyrosine kinases such as ErbB2[17,18,20,46,47]. It remains elusive whether additional pathways also control Gln-addiction, which might be targeted for therapeutic purpose. The current investigation reveals the importance of dysregulation of the Fbxo4-cyclin D1 axis, a frequent occurrence in numerous cancers[10,12,27,48], in regulating Gln-addiction independent of the known signaling pathways (Fig. 8f).

Gln is actively metabolized in tumor cells, and it is functionally involved in: (1) energy production through α-KG in tricarboxylic acid (TCA) cycle; (2) biosynthetic processes such as de novo purine and pyrimidine synthesis as well as the production of non-essential amino acids; (3) synthesis of reductive equivalents like glutathione[49]. As such, Gln metabolism is indispensable for tumor development and progression. In order to favor Gln utilization, tumor cells upregulate transporters to facilitate Gln uptake[50,51]. Indeed, we find that cells with dysregulated Fbxo4-cyclin D1 exhibit increased Gln uptake. Interestingly, although Gln uptake is elevated, energy production from OXPHOS is compromised due to mitochondrial dysfunction as the result of cyclin D1 overexpression; yet, elevated cyclin D1 drives cell cycle progression, a process that demands increased energy consumption. This paradox of energy production/consumption endows these cells with intrinsic vulnerability, which provides a therapeutic opportunity.

Glucose, another important energy source, is metabolized through glycolysis even in the presence of oxygen to replenish increased energy demand and metabolites in proliferating cells, termed Warburg effect[32]. Previous work has implicated cyclin D1-CDK4 in the regulation of glucose metabolism and gluconeogenesis through acetylation of peroxisome proliferator-activated receptor gamma coactivator 1-α (PGC-1α) mediated by GCN5[7]. Whether cyclin D1 contributes to glucose metabolism in tumor cells requires further examination.

Fbxo4 belongs to the F-box protein family that contains a conserved F-box motif[52]. Fbxo4 has several documented substrates, including cyclin D1, Trf1, and Fxr1[12,25]. Cyclin D1 degradation is precisely regulated through GSK-3β-mediated phosphorylation, which facilitates the translocation of cyclin D1 from nucleus to cytoplasm, where it is ubiquitylated by phosphorylated and oligomerized Fbxo4[10]. Loss of Fbxo4 triggers radio-resistant DNA synthesis, and sensitizes cells to chemotherapeutic intervention[12]; moreover, ectopic phospho-dead cyclin D1 (T286A) expression causes DNA re-replication, Cdt1 stabilization, DNA damage, and finally cell apoptosis[53].

These characteristics denote their potential as therapeutic targets. Consistently, we found Fbxo4 loss or cyclin D1 overexpression also drives the sensitivity of ESCC cells to another stress, Gln-depletion. Importantly, taking advantage of this genetic predisposition, targeting both GLS1 and OXPHOS effectively induces apoptosis, suppresses cell proliferation in cell culture, and inhibits the growth of tumor xenografts, demonstrating promising therapeutic efficacy. Similar observations were made using nanoparticle-formulated Bis-2-(5-phenylacetamido-1,3,4-thiadiazol-2-yl)ethyl sulfide (BPTES, another GLS1 inhibitor) and metformin in pancreatic adenocarcinoma[54], in which K-Ras is frequently mutated; however, the putative role of cyclin D1 in driving this metabolic vulnerability in pancreatic adenocarcinoma cannot be neglected.

Cyclin D1 is frequently overexpressed in human cancers and it is regarded as an oncogenic driver in the majority of these cancers[12]. Targeting the cognate catalytic CDKs of cyclin D1 is regarded as a feasible way to treat human cancers. Indeed, several CDK4/6 specific inhibitors have been developed, including PD-0332991 (palbociclib)[55], LY2835219 (abemaciclib)[56], and LEE011 (ribociclib)[57], which are being intensively investigated in human tumors in pre-clinical studies and clinical trials[12–14]. Initially, tumors respond well to CDK4/6 inhibitors; however, they will finally develop resistance that becomes a major concern and puts urgent needs to develop second-line therapies. While there are potentially numerous mechanisms of resistance, one frequent occurrence is Rb loss[16,58]. We noted that loss of Rb engenders resistance to CDK4/6 inhibitors, but it enhances the sensitivity of tumor cells to combined treatment with CB-839 and metformin. It is worth noting that one recent study highlighted CDK4/6 knockdown reprograms metabolism in tumor cells, leading to Gln-addiction, which relies on c-Myc upregulation[40]. In contrast, we demonstrate that PDR ESCC cells reprogram their metabolism to glutaminolysis in a c-Myc-independent manner. In these cells, we observed the upregulation of GLS1, the rate-limiting enzyme for glutaminolysis, directly enhances palbociclib resistance as well as Gln-addiction. These findings not only broaden our knowledge on the genetic basis of Gln-addiction, but also provide a way to treat ESCC with cyclin D1 overexpression or Rb loss (Supplementary Fig. 22), and to overcome palbociclib resistance.

mTORC1 coordinates cell growth/proliferation with the nutrient availability in proliferating cells[33,59]. As two critical nutrients, the uptake and utilization of both Gln and glucose are precisely controlled by mTORC1 in order to provide nutrients, building blocks and factors to resist stresses. Under glucose-depleted conditions, hyperactivation of mTORC1 drives AMPK activation, promotes energy consumption and reduces ATP/ADP ratio, leading to apoptosis[32]. Consistently, our data suggest activation of mTORC1, resulting from cyclin D1 dysregulation, might drive Gln-addiction through enhancing energy consumption. Recent work revealed that the mTORC1 inhibitor, MLN128, suppresses glucose metabolism, and shifts the metabolism of tumor cells to glutaminolysis in a c-Jun-dependent manner, with an elusive role of c-Myc in this process. Consistently, combined MLN128 plus CB-839 successfully overcame the metabolic reprogramming and reduced tumor burden in vivo[60]. Instead of using MLN128, metformin was combined with CB-839 and administrated in our study. Therapeutic efficacy was observed in a cyclin D1-dependent manner, hightlighting cyclin D1 as an indicator for this targeted therapy.

Owing to its activity to compromise OXPHOS, metformin is utilized to further reduce energy production in cancer cells that are already challenged by Gln-depletion or GLS1 suppression. In addition, metformin can activate AMPK and suppress mTORC1; taking these biological functions into account, the observed

therapeutic effects in our study are totally consistent with those observed with MLN128 and CB-839[60]. Furthermore, metformin can also effectively compromise glucose oxidation and reprogram metabolism to glutaminolysis[61]; the above investigation and our study mutually consolidate each other's findings. Based on these findings, our data suggest that targeting glutaminolysis and OXPHOS could be an effective therapy for ESCC with a dysregulated Fbxo4-cyclin D1 axis, and tumors resistant to palbociclib. A better molecular understanding of how Fbxo4-cyclin D1 regulates metabolic flux is currently being investigated.

In conclusion, this study elucidates the biological function of Fbxo4-cyclin D1 axis in regulating Gln-addiction through its role as a suppressor of Rb function and an activator of mTORC1. This activity facilitates the reprogramming of Gln metabolism in order to favor tumor progression. Importantly, the disruption of both glutaminolysis and mitochondrial respiration provides a promising way to treat human ESCC and to overcome palbociclib resistance.

## Methods

**MEF isolation and cell culture.** The protocol was approved by the Institutional Animal Care and Use Committee (IACUC) at the Medical University of South Carolina (MUSC ARC#: 3339); this study has complied with all relevant ethical regulations for animal testing and research. Mouse embryos (with genetic background: *Fbxo4+/+*, *Fbxo4−/−*, *Fbxo4+/+ & cyclin D1−/−* and *Fbxo4−/− & cyclin D1−/−*) were dissected out at menstrual age Day 14. The head and visceral organs and tissues were removed. Cells were maintained in MEF medium on a 3T9 passaging protocol. MEF medium contains Dulbecco modified Eagle's medium (DMEM) with 10% fetal bovine serum (FBS) (Gemini Bio-Products), 2 mM Gln, 0.1 mM non-essential amino acids, 55 μM β-mercaptoethanol, and 10 μg/ml gentamicin. NIH3T3 and HEK293T cells were purchased from the American Type Culture Collection (ATCC), and cultured in DMEM containing 10% FBS and 1% penicillin-streptomycin. TE1, TE7, TE8, TE10, and TE15 ESCC cells were kindly provided by Dr. Tetsuro Nishihara who established these cell lines[62,63], and all cells were maintained in RPMI1640 with 10% FBS and 1% penicillin-streptomycin. The earliest frozen stocks of all ESCC cell lines have been stored at the Cell Culture Core of the University of Pennsylvania. All cells were authenticated by short tandem repeat analysis for highly polymorphic microsatellites FES/FPS, vWA31, D22S417, D10S526, and D5S592 as performed by the Cell Culture Core to validate the identity of cells by comparing the earliest stocks with those grown more than 8–12 passages. TE7PDR and TE10PDR cells were established through extended culturing cells in complete medium with 1 μM palbociclib. TE7 cell, harboring cyclin D1P287A mutation, is a good model to investigate the role of cyclin D1 in regulating Gln-addiction. Morphological analysis characterizes that TE7 cells can form SCC xenografts, being consistent with previous report[64]. All used cell lines have been tested for mycoplasma contamination on a regular basis.

**Genotyping of MEFs.** MEFs were pelleted and genomic DNA was extracted with an Extract-N-Amp™ tissue PCR kit (Sigma-Aldrich). The genotyping primers were as follows: *Fbxo4*, 1loxP forward, 5′-GGCAGAGCTTGAGTTTGCAA-CATTTCAGGTG-3′, and 3loxP reverse, 5′-TCCTGATCTTT GGAAATTCTTCCTCTGAGT-3′; *cyclin D1*, Common, 5′-TAGCAGAGAGCTA-CAGACTTC G-3′, WT, 5′-CTCCGTCTTGAGCATGGCTC-3′, Mutant, 5′-CTAGTGAGACGTGCTACTTC-3′.

**Cell transfection and virus packaging.** Cells were transfected using Lipofectamine transfecting reagent (Life Technologies). For retrovirus production, pMX-puro empty vector, and vectors with Flag-tagged Fbxo4 WT, ΔN, ΔF, ΔC2, and ΔC3, and pBabe control as well as cyclin D1 WT and T286A were co-transfected with either QΨ or Ψ2 vectors; for lentivirus production, lentiviral vectors were co-transfected with pMDLg/pRRE, CMV-VSVG, and RSV-Rev vectors. The pLKO.1 shRNA constructs were purchased from Addgene: *Rb* shRNA#19 (25640), *Rb* shRNA#63 (25641), *Raptor* shRNA (1857) and *Rictor* shRNA (1853), and Dharmacon (GE Healthcare Life Sciences): TRC *GLS1* and *c-Myc* shRNAs. Virus supernatants were collected 48 and 72 h post-transfection. Infections were performed using polybrene. Thereafter, cells were either used for experiments or selected with puromycin for further analyses.

**Protein isolation and western blot analysis and quantification.** Cells were washed with 1× cold PBS and lysed in Tween 20 buffer, including 50 mM HEPES pH 8.0, 150 mM NaCl, 2.5 mM EGTA, 1 mM EDTA, 0.1% Tween 20 with protease and phosphatase inhibitor cocktail. Same amount of whole-cell extracts were resolved by sodium dodecyl sulfate polyacrylamide gel electrophoresis gel and transferred to polyvinylidene difluoride membrane. Upon blocking with 5% nonfat milk in 1× TBST, membranes were incubated with primary antibodies overnight at

4 °C. The following day, membranes were washed, and incubated with secondary antibodies for 1 h. After wash, the signals were visualized by the chemiluminescence system (PerkinElmer). Uncropped full blot images are shown in Supplementary Fig. 23. Western blot band quantification was performed using Quantity One (Bio-Rad Laboratories, Inc.). Signals were normalized to β-actin.

**Gln-depletion.** Prior to Gln-depletion, cells were seeded in complete medium (DMEM with 10% FBS and 4 mM Gln). The following day, media were replaced with Gln-free DMEM plus 10% dialyzed FBS (10 kDa cutoff) (Gemini Bio-Products). Cells cultured in Gln-free DMEM with 10% dialyzed FBS plus 4 mM Gln were used as controls.

**Gln uptake assay.** Briefly, cultured cells growing on six-well plates were washed with TS buffer (50 mM Tris-HCl and 320 mM sucrose, pH 7.4); then, cells were incubated at 37 °C for 8 min with 0.5 mCi L-2,3,4-[3H]glutamine (PerkinElmer) in DMEM (Gln-free) or 4 mM unlabeled Gln in DMEM (Gln-free) for background correction. The cells were immediately placed on ice and washed three times with ice-cold PBS; cells were lysed in 1 mM NaOH; the amount of radiolabeled Gln was determined using a Beckman Coulter LS6500 Multi-Purpose Scintillation Counter (Beckman Instruments, Fullerton, CA); the amount of [3H]glutamine taken into the cells was calculated based on the scintillation counts. Finally, [3H]glutamine uptake was normalized to protein concentrations.

**Mitochondrial membrane potential analysis.** The day before treatment, $5 \times 10^3$ MEFs and NIH3T3 cells or $1 \times 10^4$ ESCC cells were seeded into 96-well plate. The following day, cells were depleted with Gln or treated with CB-839 and/or metformin for 24 h. Control cells were treated with/without 50 μM CCCP for 20 min. All cells were incubated with 200 nM tetramethylrhodamine, ethyl ester (TMRE) for 30 min. After 1× PBS wash, fluorescent signals were monitored and recorded as RFU at wavelength (excitation: 530 nm and emission: 590 nm).

**ATP/ADP ratio analysis.** ATP/ADP ratio was analyzed using ApoSENSOR™ ADP/ATP Ratio Bioluminescent Assay Kit (BioVision). Briefly, 100 μl reaction mix in a 96-well plate was read as the background luminescence (Data A). After treatment, the cells were incubated with 50 μl nucleotide releasing buffer for 5 min. The lysate was transferred into the above 96-well plate, and 2 min later, the signals were read as Data B. Then, the samples were read again to determine the ADP levels (Data C). Finally, 1 μl ADP converting enzyme was added to the reaction to determine Data D. ATP/ADP Ratio was calculated using formula: $(Data\ B − Data\ A)/(Data\ D − Data\ C)$. The ADP/ATP ratio was normalized to protein concentrations.

**Seahorse analysis.** OCR was measured using 96-well Extracellular Flux Analyzer (XF-96) (Seahorse Biosciences). The day before analysis, $6 \times 10^3$ cells were plated; and the sensor cartridge was buffered in the calibration buffer provided by Seahorse Bioscience. The following day, cells were washed two times with phosphate buffered saline (PBS). The RS medium without Gln was added into the wells and warmed up in a 37 °C non-CO2 incubator. The injection ports of the sensor plate were filled with 10 μl of compounds or vehicles diluted in RS buffer. The sensor plate was placed into the XF-96 instrument for calibration. After calibration, the calibration fluid plate was removed and the cell plate was loaded for analysis. The measurement protocol was listed below: 2 min mix, 1 min wait, and 3 min measurement. There were totally four rate measurements upon different chemical injections (basal levels, Oligomycin, FCCP, and Antimycin/Rotenone), and each injection with four measurement cycles. OCR was finally normalized to protein concentrations.

**Mouse xenografts.** The protocol was approved by IACUC at Medical University of South Carolina (MUSC ARC#: 3339); this study has complied with all relevant ethical regulations for animal testing and research. Each group has four athymic nude, nu/nu mice (male, age: 4 weeks). TE7 ($5 \times 10^6$), TE15 ($5 \times 10^6$) and TE7PDR ($5 \times 10^6$) cells were subcutaneously injected into the flank regions. When the average volume of tumor nodules reached 60 mm³, mice were randomly assigned to four groups (day 0): vehicle, CB-839, metformin and CB-839 plus metformin. CB-839 was orally gavaged at 200 mg/kg twice per day; metformin was administrated i.p. at 250 mg/kg once per day. CB-839 was dissolved in buffer with hydroxypropyl-β-cyclodextrin (HPBCD) in 10 mmol/l citrate (pH 2, 25% (w/v)) to a final concentration of 20 g/ml (w/v). Metformin was diluted in 1 × PBS to 50 mg/ml. The treatment lasted for 20 days. Tumor size, blood glucose, and body weight were measured and recorded every 2 days. The tumor size was calculated using the formula: $0.5 \times a \times b^2$.

**Immunohistochemical (IHC) staining.** Paraffin-embedded sections were steamed, blocked, and incubated with Ki-67 (Agilent Technologies) and cleaved caspase-3 (Cell Signaling) antibodies, and signal was amplified using VECTASTAIN Elite ABC HRP Kit and detected by Vector® DAB Substrate. Thereafter, all sections were counterstained with hematoxylin, dehydrated, and mounted. Normal rabbit IgG was used as negative control. Sections were assessed, and the percentage of Ki-67-

positive cells was evaluated to compare cell proliferation and the number of cleaved caspase-3-positive cells per 2000 cells was used to compare apoptosis.

**Statistical analysis**. All measurements were taken from distinct samples, and the sample size was indicated by *n* that is listed in the relative figure legends. Plots were made either by GraphPad Prism6 or Microsoft Excel 2011. SPSS version 24.0 was used for statistical analyses. The Student *t*-test, One-way ANOVA and Two-way ANOVA were utilized to compare means. Kaplan–Meier survival curves are produced through SurvExpress Survival Analysis and the Log-Rank test was performed to compare the statistical significance. The significance level was set at 0.05 for all analyses.

**Reporting summary**. Further information on experimental design is available in the Nature Research Reporting Summary linked to this article.

**Code availability**. R-Project Bioconductor (ver. 3.4.2, 09/28/2017) was utilized to normalize the affymetrix data in order to perform the GSEA analysis. The detailed codes are listed in the Supplementary Methods part. There are no restrictions to access these codes.

## Data availability

The authors declare that all the data supporting the findings of this study are available with the article and its Supplementary Information files and from the corresponding author on reasonable request. In addition, the genome-wise data referenced during the study are available in a public repository from the NCBI Gene Expression Omnibus (https://www.ncbi.nlm.nih.gov/geo/) with GEO Accession #: GSE100942, GSE20347, GSE40513, and GSE84597, respectively.

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

## Acknowledgements

This work was supported by grants from the National Institutes of Health: P01 CA098101 (J.A.D.); and T32 DE017551 (S.Q.). We appreciate the assistance from Core Facilities and Shared Resources at the Medical University of South Carolina and Hollings Cancer Center: Flow cytometry analysis was performed in the Flow Cytometry & Cell Sorting Core and Seahorse analysis was performed in Metabolomics Core. We also thank Dr. Shikhar Mehrotra (Medical University of South Carolina) for providing methods to analyze Gln uptake.

## Author contributions

S.Q. and J.A.D.: designed the experiments. S.Q., S.P., N.O., and G.C.B.: performed experiments. S.Q., S.P., N.O., G.C.B., A.J.B., K.K.W., A.K.R., and J.A.D.: analyzed and interpreted the data. S.Q. and J.A.D.: wrote and edited the manuscript. S.Q., A.Y., G.C.B., C.C.B., and B.O.: provided reagents, materials and analysis tools.

## Additional information

**Competing interests:** The authors declare no competing interests.

