## [Peer Review File · Nature Communications]

Reviewers' Comments:

Reviewer #1:

Remarks to the Author:

Summary: The submitted manuscript interrogates how perturbations of the FBXO4/Cyclin D/RB pathway impinges on esophageal cancer models. Using matched MEFs, wild-type or FBXO4 deleted illustrated that loss of FBXO4 is associated with an increased sensitivity to glutamine withdrawal. Subsequent gene expression data illustrate that there is an association of FBXO4 status with a gene expression program of glutamine metabolism. Further data suggest that FBXO4 impact on overall metabolic activity and limits the use of oxidative metabolism and impinges on mitochondrial membrane permeability. The combination of glutamine metabolism inhibitors (CB-839) and phenformin are shown to have a modest effect on tumor growth in xenograft models. Lastly, it is shown that there are alterations in metabolic signaling that are associated with resistance to palbociclib and that such cells are more sensitive to perturbations of glutamine metabolism.

Critique: Understanding the interplay of oncogenic pathways in esophageal cancer with tumor metabolism could delineate important therapeutic vulnerabilities. While some of the data in the manuscript is provocative, there are concerns related to the interpretations of the presented data. A large degree of inconsistency across the presented immunoblots raises further concerns related to conclusions. The mechanisms through which FBXO4 acts to elicit downstream effects on metabolism remain unclear. Lastly, the degree of the proposed vulnerability on tumor growth is relatively limited.

Below a subset of the concerns are articulated:

1. There are significant concerns related to the rigor of the immunoblot data and the conclusions from them. Many blots are of marginal quality. For example, data on the requirement of Cyclin D1 is not well supported (Fig 1E—FBXO4 is variable, PARP is highly over-exposed; Fig 4F—endogenous cyclin D1? PARP under-exposed, would appear overexpression of Cyclin D1 is inducing apoptosis irrespective of Gln condition, Fig 1E--Uncleaved PARP and the Pro-caspase 3, lanes do not appear to line up). Similarly, in the analysis of the effect of palbociclib there is lack of clarity (Fig 1G-no apoptosis with Gln withdrawal). On Fig2G, why is phospho-RB induced by Gln withdrawal? The data in Fig 3B and 3E are not consistent and there is no clear basis for widely disparate results...these are just a few examples.
2. The basis of the GSEA analysis is unclear. Since esophageal cancers are growing, why would they exhibit a signature of glutamine withdrawal and how does this interface with the data presented in Fig 7? Would be advisable to demonstrate specific genes as is shown in Fig 7.
3. The interpretation of the apoptotic sensitivity in ESCC cell lines does not appear to track with FBXO4 or Cyclin D1 (Figure 2H). There is little explanation for the divergent responses to Gln withdrawal throughout the manuscript.
4. For a study focused on a metabolic vulnerability, the manuscript has little direct metabolic analysis. Are there perturbations in metabolite pools? Does loss of FBXO4 impinge on the metabolic fate of glutamine? How is glutamine or glucose utilization altered? The current study adds little to understanding these important endpoints.
5. It would appear that in specific experiments Cyclin D1 levels are diminished with Gln withdrawal, while in other experiments it is retained or induced. In Fig 4D, if Cyclin D1 is absent prior to the induction of apoptosis in FBXO4 deficient cells, how does this fit with model?
6. The therapeutic data in xenograft models shows effect of the combination therapy on tumor volume. However, this effect is very modest and it is unclear how "clinically" significant such an

effect would be. The error bars are not presented in the most appropriate fashion on the tumor growth data.

7. The data in Fig 7A are from breast cancer models, not clear how germane that would be for esophageal cancer.

8. The analysis of the PDR cell lines is interesting although a more thorough analysis would be advisable. For example, as xenografts are they in fact more sensitive to the combination treatment?

MINOR

The labels on the OCR data are unreadable (Fig 4E)

The manuscript would require significant grammatical editing

Reviewer #2:

Remarks to the Author:

The authors J. Alan Diehl and colleagues present a manuscript NCOMMS-18-05381 titled, "Targeting Glutamine-addiction and Overcoming CDK4/6 Inhibitor Resistance in Human Esophageal Squamous Cell Carcinoma." Overall, the paper is put together in a rather disjointed manner and it is difficult to follow the model they propose. I was disappointed because I think this topic of overcoming therapy resistance in aggressive cancers such as ESCC is important and the paper has promise. The manuscript would benefit greatly from a schematic. From what I did piece together of their working model it seemed to me that much of their data doesn't support their model. Many of the figures directly contradict one another for example comparing Figures 3a and 3b. The authors also do not establish a clear model of palbociclib-resistance, which seems to be the basis of the paper as indicated by the title. Rather, they seem to show that cyclin D1 overexpression is increased in all ESCC and that seems to drive gln dependency. So why include palbociclib-resistance at all? The addition of metformin to CB-839 doesn't seem to fit well with the story presented as does the inclusion of mTORC1 inhibition. These topics will require more explanation. It should also be noted that the combination of metformin + GLS inhibition has already been successfully tested in pancreatic cancer, please reference Fendt et al., Cancer Res 2013. The font on the figures is so small I had to increase the text to 500% to read it. This paper will need major revisions.

Major points:

1) Manuscript would benefit from careful copyediting, as there are a number of typos and number of sentences that don't quite make sense. Some of these have been highlighted in Minor points section.

2) The bigger problem is the fact that statements made in the manuscript are not fully supported by data presented in Figures:

a) Instead of Gln-addiction, what authors describe is activation of apoptotic signaling in response to glutamine depletion. However, there is no data on apoptotic mechanism that is proposed to be employed in response to glutamine deprivation. Furthermore, is intrinsic or extrinsic pathway involved in apoptotic response?

c) In many Figures cleaved PARP in the only read-out used to indicate apoptosis.

d) There is limited data in this manuscript that supports claims of “regulating Gln-addiction independent from the known signaling pathways”. Role of cMyc has not been adequately examined in ESCC lines used by the authors.

e) There is no data showing that uptake of glutamine is different in Fbxo4^{-/-} or CyclinD overexpressing cells.

3) What is the role of known regulators of CDK4/6 activity (for example p27/p21/p16) in response to glutamine deprivation?

4) Throughout the manuscript authors describe experiments that had conditions with and without glutamine. It would greatly improve readability of the manuscript if conditions were clearly stated in the text when conclusions are made. For example, lines 150-154 describe changes in ATP/ADP ratio, yet it is not clear if this is in glutamine rich media or glutamine deprived media.

Minor points:

Line 35: extra comma.

Line 42: E2F instead of E2f.

Line 61-63: This sentence doesn't quite make sense.

Line 64: If Fbxo4 is lost, it is already dysregulated.

Line 65-67: This sentence doesn't quite make sense.

Line 83: One would expect that glutamine re-addition after depletion would rescue cells from apoptosis. Could authors comment on their rationale for and interpretation of this assay? Comparing Figure 1A and Figure S1A it looks like there is more Cleaved Caspase 3 after glutamine re-addition to Fbxo4^{-/-} MEFs. This would suggest that both depletion and re-addition of glutamine induce apoptosis?

Line 90: “addiction” instead of “addition”.

Line 91-92: Looking at Figure S1B – levels of cMyc are not different at baseline (0 hr Gln depletion), they only look different after Glutamine re-addition. This data does not support statement in lines 91-93. Does cMyc level change upon glutamine depletion (conditions used in Figure 1A)?

Line 99: I think it makes more sense to use T286A instead of D1T286A.

Line 112: The paper doesn't discuss Head and Neck SCC, perhaps it is better to remove this data set and focus on ESCC.

Line 123-124: In order to state that apoptosis is “obviously increased” in particular cell lines, it would be important to show additional markers of apoptosis, rather than single cleaved PARP blot.

Line 136: Please list NES scores and FDR q values for GSEA data in Figure or Figure Legends.

Line 137: When comparing basal phosphorylation of some mTORC1 targets, one can make an argument (based on Figure 3A) that overexpression of Cyclin D1 T286A results in increased phosphorylation of p70S6K1, pS6 and p4EBP1, but from the Figure 3A the same can't be said

about overexpression of wild type Cyclin D.

Line 138: Do ESCC cell lines (TE7, TE8, TE10, TE15, TE1) have mutations in any other oncogenes or tumor suppressors?

Would it be possible to list genotype of each cell line on the figure itself? There are a different cell lines used on many blots in the same figure, it would be helpful to have genotypes for each cell line listed.

Figure 3A and 3B: from the figure legend it seems that Cyclin D1 and Cyclin D1 T286A were overexpressed in both NIH3T3 and ESCC cell lines? Is this correct?

Line 142: What cell line was used in Figure 3C?

Line 142: From Figure S5C and S5D it is really not possible to determine if Rapamycin suppressed cleaved PARP or cleaved caspase 3.

Line 157: What was the rationale for using Asparagine in rescue experiments?

Line 158: Figure S6C does not have Gln positive control.

Line 161: Figure 4E: Legend on this figure is not legible.

Line 167: "Correlates" instead of "leads to".

Line 172-174: Only metformin is used to treat type II diabetes, while phenformin is not FDA approved.

Line 174: What were concentrations of CB-839, Phenformin and Metformin used in Figures 5A-5I?

Line 175-179: Combination therapy does not appear to induce significantly increased "apoptotic rate" when Cyclin D (WT or T286A) is overexpressed – cleaved PARP is not different in lanes with single CB-839 compared to CB-839+Metformin (Figure 5D). Blots showing cleaved PARP in Figure S7 are underexposed thus making any interpretation very hard.

Line 224-225: From the Figure S12B, only TE7 cells showed increased cleaved PARP levels in response to Gln depletion (comparing TE7 Parental to PDR), but not TE10 cells. Please revise text to more clearly describe data shown in Figures.

Line 227: If it is impossible to establish resistant cells (resistant to what?) what type of cells were described on line 222-223?

Reviewer #3:

Remarks to the Author:

In this study Qie et al. demonstrated that esophageal squamous cancer cells with dysregulated cyclin D1-CDK4/6 either due to genetic mutations or loss of Fbxo4, an E3 ubiquitin ligase for the cyclin complex present Glutamine-addiction. The authors further showed that dysregulation of Fbxo4-cyclin D1 axis leads to mitochondrial dysfunction and metabolic reprogramming. At the molecular level the authors found that Rb and mTORC1 contribute to Glutamine-addiction upon the dysregulation of the Fbxo4-cyclin D1 axis. Rb loss is accompanied by increased levels of glutaminase 1 (GLS1) which promotes Gln-addiction. More importantly the authors showed that combined treatment with CB-839, a GLS1 inhibitor and metformin/phenformin effectively induces cell apoptosis and suppresses cell proliferation in vitro, in xenografts and in palbociclib resistant

tumors. These findings are novel and provide important insights into the ESCC vulnerability where cyclin D1-cdk4/6 abnormalities are common. The experiments were well designed and performed, and the manuscript was well written. That said this reviewer has some concerns that require further experiments or discussion.

1. Validate changes in the levels of some putative downstream target of the mTOR pathway in ESCC cells at protein or transcript level after inhibitor treatment.
2. What is the level of GIs in ESCC biopsies?

The authors would like to express their sincere appreciation to the editor and the reviewers for their attentive work, and their constructive and positive comments. The following are the detailed replies to Reviewers' questions in a Point-by-Point way:

Reply to Reviewer #1:

Summary: The submitted manuscript interrogates how perturbations of the FBXO4/Cyclin D/RB pathway impinges on esophageal cancer models. Using matched MEFs, wild-type or FBXO4 deleted illustrated that loss of FBXO4 is associated with an increased sensitivity to glutamine withdrawal. Subsequent gene expression data illustrate that there is an association of FBXO4 status with a gene expression program of glutamine metabolism. Further data suggest that FBXO4 impact on overall metabolic activity and limits the use of oxidative metabolism and impinges on mitochondrial membrane permeability. The combination of glutamine metabolism inhibitors (CB-839) and phenformin are shown to have a modest effect on tumor growth in xenograft models. Lastly, it is shown that there are alterations in metabolic signaling that are associated with resistance to palbociclib and that such cells are more sensitive to perturbations of glutamine metabolism.

Reply: Thank you for your positive comments on our study. For the suppressing effects of combined CB-839 and metformin on ESCC xenografts, if comparing control groups and treated groups, we did observe statistical significance in TE7 but not in TE15 xenografts, and this observation is consistent with our experimental findings in cell culture. Furthermore, we now include a rationale describing why utilization of CB-839 has limited impact in vivo, and how we are going to improve the therapeutic efficacy in the future; **please refer to our reply to Question #6.**

Critique: Understanding the interplay of oncogenic pathways in esophageal cancer with tumor metabolism could delineate important therapeutic vulnerabilities.

Reply: Thank you for the positive assessment on the importance of our study.

While some of the data in the manuscript is provocative, there are concerns related to the interpretations of the presented data. A large degree of inconsistency across the presented immunoblots raises further concerns related to conclusions.

Reply: As described below, the concerns mentioned reflect issues with data presentation. With respect to the reviewers' comments, we have made significant changes to our description and included new data as well as longer exposures of western blots as requested yet still insuring they are representative data (Please also refer to our reply to Question #1).

The mechanisms through which FBXO4 acts to elicit downstream effects on metabolism remain unclear.

Reply: While there remain observations that warrant ongoing investigation, our data clearly demonstrate that one aspect of this mechanism reflects dysregulation of cyclin D1 and increased cyclin D1/CDK4 kinase activity, leading to mitochondrial dysfunction. This hypothesis is supported by data from cancer cell lines, reconstituted fibroblasts and cells with rescue wherein cyclin D1 is knocked out, CDKs are pharmacologically suppressed, or Rb is lost. Our data reveal that this results in increased glutaminolysis through GLS1 upregulation as a clear mechanism.

Lastly, the degree of the proposed vulnerability on tumor growth is relatively limited.

Reply: We respectfully disagree with this suggestion, because the impact on TE7 xenografts with

dysregulated cyclin D1 is highly significant. That fact that the impact on TE15 xenografts is limited reflects that this cell line harbors wild type cyclin D1 and wild type Fbxo4. Importantly, these are the expected results based upon our mechanistic analyses, which provide rationale to selectively use this therapeutic strategy according to cyclin D1 gene status. Importantly, our new data suggest combined treatment demonstrate much stronger effects on **suppressing TE7 PDR (palbociclib resistant) xenografts (Fig. 8), still being consistent with our hypothesis.**

Response to specific concerns:

1. There are significant concerns related to the rigor of the immunoblot data and the conclusions from them. Many blots are of marginal quality. For example, data on the requirement of Cyclin D1 is not well supported (Fig 1E—FBXO4 is variable, PARP is highly over-exposed;

Reply: in Fig. 1e, we utilized independent MEF isolates to test our hypothesis that “cyclin D1 is one of the downstream factors that contributes to Fbxo4 loss-mediated Gln-addiction”. As such, independent MEF cells may have variation of Fbxo4 levels, but cyclin D1 levels are consistently higher in Fbxo4^{-/-}, cyclin D1^{+/+} than Fbxo4^{+/+}, cyclin D1^{+/+} cells. However, the variations don't interfere with the interpretation of our data and supporting our conclusions. Importantly, PARP blot is overexposed to permit clear observation of cleaved PARP, the indicator of cell apoptosis; therefore, both PARP and cleaved PARP can be shown on the same blot. In order to address reviewer's concern, we included a short exposure PARP from the same blot in Fig. 1e. Fig 1F—endogenous cyclin D1? PARP under-exposed, would appear overexpression of Cyclin D1 is inducing apoptosis irrespective of Gln condition,

Reply: To make this clearly, we have re-run the same samples and blotted for cyclin D1, PARP and β -actin (Fig. 1f). Both endogenous and ectopic cyclin D1 are demonstrated in the same blot (indicated by arrows); the lower band is endogenous cyclin D1.

For the PARP blot, in order to show both full length PARP and cleaved PARP on the same blot, we chose this blot with current exposure. Importantly, in WT D1 group, we did observe more PARP cleavage when comparing Gln-depleted group with control group; please remember we also included cleaved caspase-3 as another apoptotic indicator that clearly demonstrates Gln-depletion induces more apoptosis in cells with WT D1; consistently, Annexin V staining also support more cell apoptosis is induced in cells with WT D1 (Supplementary Fig. 2b).

We agree that both cyclin D1 and D1T286A are pro-apoptotic as the reviewer pointed out. This phenomenon was noted in the early 1990s (Quelle et al Genes Dev 1993). My lab has reported the mechanisms how overexpressed, nuclear cyclin D1/CDK4 triggers cells apoptosis (reported by us in “Cancer Discov. 2015 Mar;5(3):288-303.”). Importantly, this doesn't take away from our data that demonstrate esophageal cancer cells with cyclin D1 accumulation (due to Fbxo4 mutation) or cyclin D1 stabilizing mutation (D1P287A; Benzeno et al Oncogene 2006) sensitizes cells to Gln withdrawal, and overexpression of D1 or D1T286A in non-tumorigenic murine fibroblasts confers sensitivity to Gln withdrawal.

Fig 1E--Uncleaved PARP and the Pro-caspase 3, lanes do not appear to line up.

Reply: They do indeed align, but one issue is that the expression of pro-caspase-3 is low in the last MEF line. To provide clarity, we have included a blot with a longer exposure of pro-caspase-3 (Fig. 1e). In addition, the original blot data are also provided (Supplementary Fig. 23).

Similarly, in the analysis of the effect of palbociclib there is lack of clarity (Fig 1G-no apoptosis with Gln withdrawal).

Reply: The concern for PARP blots in Fig. 1g is analogous to Fig. 1f (exposure time). Herein, we took the same sample and have repeated PARP blot. Clearly, we demonstrate cell apoptosis indicated by PARP cleavage. In addition, we provide FACS to demonstrate the percentage of Annexin V positive cells (Supplementary Fig. 2b).

On Fig 2G, why is phospho-RB induced by Gln withdrawal?

Reply: TE1 cells lack Rb. Thus in Fig. 2g, this exact band in TE1 cells without Gln, which migrates faster than the phospho-Rb, is a non-specific band. Moreover, the blot for phospho-Rb and blot for total Rb are from the same gel; this same membrane was stripped and redeveloped for total Rb just after blotting with phospho-Rb (Please also refer to the original blots in Supplementary Fig. 23). We use an “Arrow” to point out the correct phospho-Rb band and “Open triangle” to indicate the non-specific band for clarity.

The data in Fig 3B and 3E are not consistent and there is no clear basis for widely disparate results

Reply: In fact, Gln-depletion times are totally different for these two figures, which may cause confusion that the variation of blot results presents. To make it clear, Fig. 3b is intended to show basal mTORC1 activity after 0, 0.5, 1, 2 and 4 h Gln-depletion, while Fig. 3e is used to demonstrate cell apoptosis upon 24 h Gln-depletion. These two figures are included to answer different questions. In order to support our hypothesis, we think these results make sense. We now add our rationale why these experiments were done and the differences between these two experiments in the main text.

2. The basis of the GSEA analysis is unclear. Since esophageal cancers are growing, why would they exhibit a signature of glutamine withdrawal and how does this interface with the data presented in Fig 7? Would be advisable to demonstrate specific genes as is shown in Fig 7.

Reply: Our intention for GSEA analysis is to indicate the dysregulation of cell cycle genes and the genes related to Gln-depletion in order to provide rationale to choose ESCC as our model. ESCC is a solid tumor and it frequently has ineffective blood supplying, which may lead to nutrient withdrawal; therefore, we checked genes related to Gln-depletion. Moreover, the signature genes of Gln withdrawal used in our study was published before (Mol Cell Biol. 2002 Aug;22(15):5575-84.); importantly, this signature is broadly used and well recognized, and it helps us getting an unbiased big picture of Gln metabolic status in ESCC relative to normal tissues.

Following the reviewer’s advice, we also included a detailed profiling to show the genetic alterations of Gln metabolism genes (Please refer to Supplementary Fig. 4).

3. The interpretation of the apoptotic sensitivity in ESCC cell lines does not appear to track with FBXO4 or Cyclin D1 (Figure 2H). There is little explanation for the divergent responses to Gln withdrawal throughout the manuscript.

Reply: In this study, the role of “Fbxo4-cyclin D1” signaling pathway in Gln-addiction was investigated. Our hypothesis is that dysregulation of “Fbxo4-cyclin D1”, as well as downstream factor “Rb”, will drive cellular dependency on Gln. This “dysregulation” doesn’t merely mean

protein levels; it also emphasizes on the activity of a specific factor (now we directly list the gene status of different cells in relative figures).

For example, Fbxo4 loss-of-function mutation leads to cyclin D1 accumulation and finally, Gln-addiction (like TE10 cells with Fbxo4 S8R mutation). Loss of Rb (like TE1 cells) also enhances Gln-addiction even with low cyclin D1 levels, which is consistent with previous reports (J Cell Biol 125, 625-638 and PNAS 91, 2945-2949); because of Rb is one critical downstream factor of cyclin D1-CDK4 and its loss drive cell cycle progression in a cyclin D1-independent manner. Therefore, due to genetic variations of different ESCC cells, variations of apoptosis were monitored in different cells upon Gln-depletion. Generally and consistently, cells with Fbxo4 mutation (which dysregulates cyclin D1), cyclin D1 mutation or Rb loss show more cell apoptosis upon Gln withdrawal.

Although cyclin D1 can partially regulate cell apoptosis, it doesn't account for all in the absence of Fbxo4. In order to support our hypothesis, we also tested c-Myc levels and found it doesn't attribute to Gln-addiction in cells with dysregulated "Fbxo4-cyclin D1" axis (Supplementary Fig. 6). It is worth noting that given heterogeneity of tumor lines, it is not unexpected to observe some variation in phenotypes that is why we also utilize fibroblasts in which this pathway was altered as proof of principle. Taken together, our data are consistent in that cell lines with dysregulated "Fbxo4-cyclin D1" are more sensitive to Gln withdrawal than those maintaining wild type signaling.

4. For a study focused on a metabolic vulnerability, the manuscript has little direct metabolic analysis. Are there perturbations in metabolite pools? Does loss of FBXO4 impinge on the metabolic fate of glutamine? How is glutamine or glucose utilization altered? The current study adds little to understanding these important endpoints.

Reply: We totally agree with the reviewer for this question and it will be one of the major questions we are going to answer in the future. But for now, this manuscript focuses more on developing a new therapeutic strategy other than a pure metabolism investigation. In this study, we included some necessary metabolic analyses in order to support our hypothesis, such as, [³H]Gln uptake assay to show the changing of Gln intake, and NMR analysis to demonstrate the GLS1 inhibitor alters Gln and glutamate levels in mouse xenografts. For long-term goal, we are going to dig more into the alteration of metabolic pool for both glucose and Gln in cells with dysregulated "Fbxo4-cyclin D1" axis. To make it clear, we added some information in the Discussion part.

5. It would appear that in specific experiments Cyclin D1 levels are diminished with Gln withdrawal, while in other experiments it is retained or induced. In Fig 4D, if Cyclin D1 is absent prior to the induction of apoptosis in FBXO4 deficient cells, how does this fit with model?

Reply: This is an excellent point that we would like to clarify. Base on data from our lab and other groups, Gln-depletion compromises cyclin D1 expression. For a specific figure like Fig. 1e, we run the cyclin D1 blot from cells cultured in complete medium with Gln in order to demonstrate genotypes of different MEF cells. The above is the reason why cyclin D1 is detected in Fig. 1e. We now add the condition we used and the rationale why we did it this way in the figure legend.

For Fig. 4d, we agree that cyclin D1 levels are reduced in both Fbxo4 +/+ and -/- MEFs; however, from the blot one can still tell that more cyclin D1 is detected in Fbxo4-/- MEFs than

in $+/+$ counterparts. Our conclusion here is that more residual cyclin D1 drives apoptosis in $Fbxo4^{-/-}$ MEFs.

6. The therapeutic data in xenograft models shows effect of the combination therapy on tumor volume. However, this effect is very modest and it is unclear how “clinically” significant such an effect would be. The error bars are not presented in the most appropriate fashion on the tumor growth data.

Reply: For therapeutic effects, if comparing control and treated xenografts, we did see statistical significance in TE7 but not in TE15 xenografts, which supports our hypothesis tumors with dysregulation of “Fbxo4-cyclin D1” are more sensitive to the combined treatment. As for the modest effects the reviewer mentioned, there are two major reasons:

(I) The efficiency of delivering CB-839 into xenografts. A previous report (Proc Natl Acad Sci U S A. 2016 Sep 6;113(36):E5328-36.) suggests that chemical properties of CB-839 and BPTES (another GLS1 inhibitor) determines a bad delivering efficiency; therefore, they used nanoparticle-encapsulated BPTES, which obviously enhances the delivering efficiency. For our future studies, we are also going to encapsulate CB-839 with nanoparticles in order to enhance the delivering efficiency.

(II) The growing property of ESCC xenografts themselves. Based on our experience, the growth rate of ESCC xenografts is not as fast as tumors like pancreatic cancer and melanoma. This intrinsic characteristic also makes the suppressing effects less dramatic than other types of cancers.

In addition, we have now included a figure to test the suppressing effects of combined treatment on TE7 PDR (palbociclib resistant) xenografts. In this Fig 8a, one can readily tell that combined treatment has a much stronger therapeutic effect.

7. The data in Fig 7A are from breast cancer models, not clear how germane that would be for esophageal cancer.

Reply: Good point. We performed the analysis in Fig. 7a to determine whether palbociclib-resistant tumors show metabolic vulnerability given the prediction of dysregulated cyclin D1/CDK4-Rb signaling as a consequence of acquired resistance to palbociclib. Our rationale for this is listed below: taking the advantage of the online available GEO datasets, we screened genes that are related to Gln metabolism after treatment with palbociclib, which can give us a quick and clear clue if our hypothesis is correct. To correlate these findings with our investigation, we establish TE7 and TE10 cells that were resistant to palbociclib (Supplementary Fig. 18), and consistently, we observed increased GLS1 mRNA and protein levels (Fig. 7b&c) like that for breast cancer cells in Fig. 7a. Taken together, these data not only consolidate our hypothesis, but also suggest the metabolic alterations are a common phenomenon that is not only limited to one cancer type.

8. The analysis of the PDR cell lines is interesting although a more thorough analysis would be advisable. For example, as xenografts are they in fact more sensitive to the combination treatment?

Reply: We thank the reviewer for this suggestion. Per the reviewer’s suggestion, we injected TE7 PDR cells into nude mice and established TE7 PDR xenografts. Thereafter, the same combined treatment and vehicle control were administrated to these mice. Not surprisingly, we observed stronger suppressing effects in these xenografts than their parental TE7 xenografts. In addition,

IHC staining also revealed more decreased cell proliferation and increased cell apoptosis (Fig. 8 and Supplementary Fig. 21). To our knowledge, this is the first study that found efficient therapeutic effects on palbociclib resistant tumors through combining CB-839 and metformin together. These data not only support our hypothesis, but also provide mechanistic basis how to overcome palbociclib resistance.

MINOR

The labels on the OCR data are unreadable (Fig 4E)

Reply: We have changed the font size for the labels in this figure.

The manuscript would require significant grammatical editing

Reply: We have carefully edited the manuscript.

Reply to Reviewer #2:

The authors J. Alan Diehl and colleagues present a manuscript NCOMMS-18-05381 titled, “Targeting Glutamine-addiction and Overcoming CDK4/6 Inhibitor Resistance in Human Esophageal Squamous Cell Carcinoma.” Overall, the paper is put together in a rather disjointed manner and it is difficult to follow the model they propose. I was disappointed because I think this topic of overcoming therapy resistance in aggressive cancers such as ESCC is important and the paper has promise. The manuscript would benefit greatly from a schematic.

Reply: Thank you for your suggestion. We improved the proposed model (Fig. 8f) to summarize the major findings in this paper. Briefly, this study started from why and how dysregulated “Fbxo4-cyclin D1” axis drives Gln-addiction. Due to the intrinsic vulnerability of ESCC cells, combined treatment (CB-839 and metformin) was successfully utilized to control tumor growth through inducing cell apoptosis and suppressing cell proliferation. Lastly, based on the dysregulation of cyclin D1-Rb and the presence of reprogramming of Gln metabolism in palbociclib-resistant cells, we reasoned that these cells should be highly sensitive to treatment through targeting of glutaminolysis and oxidative phosphorylation. From the above, one can observe that overcoming palbociclib resistance is one aspect that we are focusing on and have provided additional experimental evidence in this revised manuscript. Given the lack of therapeutic improvements in esophageal cancer, we think our data reveal a promising future how patients will be benefited from this study, and importantly, it also broadens our knowledge on the molecular mechanism how Gln-addiction is regulated. **We have included the schematic in figure legend of Fig. 8f.**

From what I did piece together of their working model it seemed to me that much of their data doesn't support their model. Many of the figures directly contradict one another for example comparing Figures 3a and 3b.

Reply: **This conclusion may be due to our failure to provide a clear presentation and description of the experiments and the data in these two figures.** Therefore, we respectfully disagree with this comment, and we think our data are entirely consistent with our hypothesis “dysregulated Fbxo4-cyclin D1 axis increases Gln-addiction”. Furthermore, our data are also consistent with previous reports: Rb loss contributes to Gln-dependency (Genes Dev. 2013 Jan 15;27(2):182-96. & Oncogene. 2014 Jan 30;33(5):556-66.).

With regard to perceived inconsistency, in Fig. 3a, NIH3T3 cells with *ectopic cyclin D1* were used, while in Fig. 3b, ESCC cells (TE7, TE10 and TE15) were used. In both models, cyclin D1 levels are important for mTORC1 activation. However, in Fig. 3a the expression of WT D1 is lower than D1T286A (given that D1T286A is highly stable through resistant to ubiquitylation-dependent proteolysis), which is why WT D1 doesn't promote mTORC1 activation as effectively as D1T286A does in NIH3T3 cells. Indeed, D1T286A expressing cells are more sensitive to Gln-depletion than WT D1 overexpressing cells. Likewise, in Fig. 3b, TE7 and TE10 cells have more cyclin D1 than TE15 cells, which is why mTORC1 is highly activated in TE7 and TE10 cells comparing to TE15 cells. Taken together, these data in Fig. 3a and 3b are consistent and support each other very well.

The authors also do not establish a clear model of palbociclib-resistance, which seems to be the basis of the paper as indicated by the title.

Reply: We apologize for not providing additional data to demonstrate these cells are palbociclib resistant. To address the reviewer's concern, we have included data to support the cells are palbociclib resistant (Supplementary Fig. 18). We included a description of the establishment and characterization of cells resistant to palbociclib. In addition, we also cited our previous paper, which provides more information on establishing these palbociclib resistant cells.

The addition of metformin to CB-839 doesn't seem to fit well with the story presented as does the inclusion of mTORC1 inhibition. These topics will require more explanation.

Reply: In order to support our hypothesis in this manuscript, metformin was utilized as an inhibitor of oxidative phosphorylation that can further disrupt the already existing energy stress in cells with dysregulated "Fbxo4-cyclin D1" axis. As stated, our data suggest this dysregulation promotes the cells' dependency on oxidative phosphorylation; therefore, in order to promote the therapeutic efficiency, metformin was combined with Gln-depletion or GLS1 inhibition to further drive the energy catastrophe and kill tumor cells. Consistently, a recent paper (Cancer Cell. 2018 May 14;33(5):905-921.) also found mTORC1 inhibitor plus GLS1 inhibitor successfully alleviates tumor burden in a transgenic mouse model with spontaneous development of lung squamous cell carcinoma. Following the reviewers' comments, we add discussion on the role of metformin as mTORC1 inhibitor and how these findings in this paper are fitting well with our knowledge in this area.

It should also be noted that the combination of metformin + GLS inhibition has already been successfully tested in pancreatic cancer, please reference Fendt et al., Cancer Res 2013.

Reply: Thank you for your advice. In the last submission, we have already cited this paper (Proc Natl Acad Sci U S A. 2016 Sep 6;113(36):E5328-36.). In this PNAS paper, the authors focused on developing new treatment by combining BPTES and metformin in order to treat pancreatic cancers; however, their whole story only focused on metabolism without elucidating the genetic mutations that drives tumor cells' vulnerability. When it comes to our paper, one can easily tell that we do bring new knowledge to this area: first, we found and established the molecular basis, in which dysregulation of "Fbxo4-cyclin D1" axis drives Gln-addiction through inactivating Rb and activating mTORC1 that lead to metabolism reprogramming; second, using ESCC as our model, we found combined CB-839 and metformin treatment can effectively suppress tumor growth through reducing cell proliferation as well as inducing cell apoptosis; last but importantly, this combined treatment can be used to overcome palbociclib resistance, and it provides a promising way as an alternative second line treatment.

In this paper “Fendt et al., Cancer Res 2013”, the authors characterized the metabolism reprogramming upon metformin treatment that alters glucose metabolism, and shifts tumor cells’ metabolism to Gln catabolism. This paper and our manuscript mutually support each other, and importantly, the “Cancer Res” paper reveals the metabolic basis why combined treatment has therapeutic efficacy. We have cited this paper to support our hypothesis in the Discussion part.

The font on the figures is so small I had to increase the text to 500% to read it. This paper will need major revisions.

Reply: Thank you for this suggestion. We have changed the font size wherever it is necessary.

Major points:

1) Manuscript would benefit from careful copyediting, as there are a number of typos and number of sentences that don’t quite make sense. Some of these have been highlighted in Minor points section.

Reply: We carefully went through the whole manuscript and made changes wherever necessary.

2) The bigger problem is the fact that statements made in the manuscript are not fully supported by data presented in Figures:

Reply: We respectively disagree with the reviewer’s comments. Our detailed explanations are listed below in a Point-by-Point manner.

a) Instead of Gln-addiction, what authors describe is activation of apoptotic signaling in response to glutamine depletion. However, there is no data on apoptotic mechanism that is proposed to be employed in response to glutamine deprivation. Furthermore, is intrinsic or extrinsic pathway involved in apoptotic response?

Reply: We now provided a clear statement to describe Gln-addiction or Gln-dependency, which emphasizes both survival and proliferation of tumor cells depends on Gln and cell apoptosis is an important readout for Gln-addiction. To somehow, we agree that it is important to dissect the mechanisms how apoptosis is induced upon Gln-depletion; however, due to the limitations of this manuscript, we didn’t include this topic, instead we put more focus on developing new therapy based on the metabolic vulnerability of tumor cells, and we think at the beginning of a project, it is more critical to answer questions with clinical importance.

In addition, how or whether it is intrinsic or extrinsic apoptosis is not the novel aspect of our study, and increased data along this line would take away from the impact that dysregulated “Fbxo4-cyclin D1” axis sensitizes cells to Gln withdrawal, which is in fact clearly defined and broadly accepted as Gln-addiction or Gln-dependency (eg. Wise et al PNAS 2008).

c) In many Figures cleaved PARP in the only read-out used to indicate apoptosis.

Reply: We agree that we mainly used cleaved PARP as an indicator for apoptosis; however, in some critical figures, we also detected apoptosis indicated by FACS for Annexin V-positive cells. To address the reviewer’s concern, we performed additional FACS to demonstrate cell apoptosis following Annexin V staining. Critically, all experimental approaches are consistent with PARP cleavage and strongly support our hypothesis.

d) There is limited data in this manuscript that supports claims of “regulating Gln-addiction independent form the known signaling pathways”. Role of cMyc has not been adequately

examined in ESCC lines used by the authors.

Reply: Thank you for this suggestion. Accordingly, we evaluated PARP cleavage and Annexin V staining after c-Myc knockdown in TE7 and TE10 cells. Western blot analysis demonstrates that c-Myc is effectively knocked down; however, apoptosis is not compromised, supporting c-Myc is not a direct driver of Gln-addiction in cells with “Fbxo4-cyclin D1” dysregulation (Supplementary Fig. 6). In addition, c-Myc knockdown also doesn't compromise GLS1 expression in TE7 and TE10 cells resistant to palbociclib, highlighting other mechanisms might promote GLS1 upregulation in these cells. These findings are consistent with a previous study “the effects of c-Myc on glutamine metabolism are context dependent” in *Cell Metab.* 2012 Feb 8;15(2):157-70.

e) There is no data showing that uptake of glutamine is different in Fbxo4^{-/-} or CyclinD overexpressing cells.

Reply: This is an excellent suggestion and we now provide data demonstrating increased [³H]Gln uptake in cells with either Fbxo4 loss or cyclin D1 overexpression (Supplementary Fig. 1a, 2c and 7a), being consistent with Gln-addiction in these cells. In addition, we also found that palbociclib-resistant cells have a higher [³H]Gln uptake compared to their parental counterparts (Fig. 7d), supporting increased sensitivity of these cells to combined treatment.

3) What is the role of known regulators of CDK4/6 activity (for example p27/p21/p16) in response to glutamine deprivation?

Reply: This is an important question and it is also a critical direction of our future studies. By checking the literature, there is one report on p21 and Gln-depletion (*Oncogene.* 2017 Apr 6;36(14):1991-2001.). Gln-depletion induces p21 upregulation; while knockdown of p21 drives cell apoptosis upon Gln withdrawal, indicating its protective role under this stress condition. In general, we suggest that the role of p21/p27 (as activators of cyclin D1-CDK4) and p16 (as an inhibitor of cyclin D1-CDK4) is beyond the scope of this current study, particularly given that p16 levels do not result in a basal increase in cyclin D1-CDK4 activity, and D1T286A is equally responsive to Ink and CIP/KIP family members (Diehl et al *Genes Dev* 1997; 1998). Finally, p27 and p21 have activities independent of CDK regulation making such analysis quite complex and thus would warrant significant investigation in the future.

4) Throughout the manuscript authors describe experiments that had conditions with and without glutamine. It would great improve readability of the manuscript if conditions were clearly stated in the text when conclusions are made. For example, lines 150-154 describe changes in ATP/ADP ratio, yet it is not clear if this is in glutamine rich media or glutamine deprived media.

Reply: We now clearly indicate the conditions we used either in the main text or in figure legends. For the ATP/ADP ratio analysis, we used complete medium with Gln, and this statement has been included in the main text.

Minor points:

Line 35: extra comma.

Reply: We deleted the extra comma.

Line 42: E2F instead of E2f.

Reply: E2f is actually a correct nomenclature that is used interchangeably to E2F. However, we

have made the suggested change.

Line 61-63: This sentence doesn't quite make sense.

Reply: This sentence has been modified.

Line 64: If Fbxo4 is lost, it is already dysregulated.

Reply: We deleted "dysregulated".

Line 65-67: This sentence doesn't quite make sense.

Reply: We modified the sentence for clarity.

Line 83: One would expect that glutamine re-addition after depletion would rescue cells from apoptosis. Could authors comment on their rationale for and interpretation of this assay? Comparing Figure 1A and Figure S1A it looks like there is more Cleaved Caspase 3 after glutamine re-addition to Fbxo4^{-/-} MEFs. This would suggest that both depletion and re-addition of glutamine induce apoptosis?

Reply: A good point. Actually, we also noted these interesting data at the very beginning and we are still working on the mechanism how cell apoptosis is induced at early timepoints upon Gln re-addition. Possible mechanisms include: the production of reactive oxygen species, like cell damage induced by ischemia and reperfusion; or it takes time for re-added Gln to suppress cell apoptosis, which is supported by the fact that at later timepoints, cell apoptosis is obviously suppressed.

Line 90: "addiction" instead of "addition".

Reply: We changed "addiction" to "addition".

Line 91-92: Looking at Figure S1B – levels of cMyc are not different at baseline (0 hr Gln depletion), they only look different after Glutamine re-addition. This data does not support statement in lines 91-93. Does cMyc level change upon glutamine depletion (conditions used in Figure 1A)?

Reply: We have collected new samples and repeated the western blots in order to determine c-Myc levels upon Gln withdrawal. Consistent with previous data, new western blots demonstrate c-Myc levels are lower in Fbxo4^{-/-} MEFs than those in +/+ MEFs (Supplementary Fig. 1c). In addition, we have also included data to support c-Myc is not the direct driver of Gln-addiction in cells with dysregulated "Fbxo4-cyclin D1" axis (Please also refer to Question 2-d).

Line 112: The paper doesn't discuss Head and Neck SCC, perhaps it is better to remove this data set and focus on ESCC.

Reply: The reason we included Head and Neck SCC is to make our argument stronger, in which the reprogramming of Gln metabolism is a common phenomenon that is not only limited to ESCC; furthermore, these data also provides molecular basis to indicate a probably wide potential of using this combined treatment in other tumors.

Line 123-124: In order to state that apoptosis is "obviously increased" in particular cell lines, it would be important to show additional markers of apoptosis, rather than single cleaved PARP blot.

Reply: We have added FACS for Annexin V staining to indicate cell apoptosis (Supplementary Fig. 7c), which is consistent with PARP cleavage.

Line 136: Please list NES scores and FDR q values for GSEA data in Figure or Figure Legends.

Reply: For the original submission, we did include these statistical data in the relative supplementary tables. However, to make it clear, we have put the statistical results in relative figure legends.

Line 137: When comparing basal phosphorylation of some mTORC1 targets, one can make an argument (based on Figure 3A) that overexpression of Cyclin D1 T286A results in increased phosphorylation of p70S6K1, pS6 and p4EBP1, but from the Figure 3A the same can't be said about overexpression of wild type Cyclin D.

Reply: This is a good point. For the overexpression experiments, one can tell that D1T286A protein levels are higher than WT cyclin D1. According to our hypothesis cyclin D1 levels are important for mTORC1 activation; therefore, it is not surprising to observe more mTORC1 activation in D1T286A cells than cells with WT cyclin D1, and this mTORC1 activation correlates with more cell apoptosis in D1T286A cells than that with WT D1 upon Gln-depletion.

Line 138: Do ESCC cell lines (TE7, TE8, TE10, TE15, TE1) have mutations in any other oncogenes or tumor suppressors?

Reply: Based on our screening, we didn't find other known mutations involved in Gln-addiction.

Would it be possible to list genotype of each cell line on the figure itself? There are a different cell lines used on many blots in the same figure, it would be helpful to have genotypes for each cell line listed.

Reply: Good point, we have labeled the genotypes of ESCC cells wherever it is necessary in relative figures.

Figure 3A and 3B: from the figure legend it seems that Cyclin D1 and Cyclin D1 T286A were overexpressed in both NIH3T3 and ESCC cell lines? Is this correct?

Reply: No. Actually, we have two different panels: Fig. 3a is NIH3T3 cells with WT and T286A D1 overexpression; Fig. 3b is ESCC cells (TE7, TE10 and TE15) without ectopic cyclin D1 expression. Now, we correct the misleading figure legends.

Line 142: What cell line was used in Figure 3C?

Reply: NIH3T3 cells are used in Fig. 3c. We have emphasized this information in the figure legend.

Line 142: From Figure S5C and S5D it is really not possible to determine if Rapamycin suppressed cleaved PARP or cleaved caspase 3.

Reply: In Fig. S5c (now it is Supplementary Fig. 8a), we quantified the western blots for both cleaved PARP and cleaved caspase-3, and the quantification results support that rapamycin does suppress their cleavage. In addition, we re-run the blots for Fig. S5d (now it is Supplementary Fig. 8b). Obviously, Rad001, another mTORC1 inhibitor, successfully inhibited cell apoptosis induce by Gln-depletion. The numbers under the blots indicate the relative signal quantification

of cleaved PARP and cleaved caspase-3, from which one can tell the inhibiting effects very easily.

Line 157: What was the rationale for using Asparagine in rescue experiments?

Reply: Asparagine has been shown to rescue cell apoptosis induced by Gln-withdrawal (Mol Cell. 2014 Oct 23;56(2):205-218.). Asparagine is used as an alternative nitrogen source under Gln-depleted conditions. In order to dissect why Gln is important for cell survival, we tested the rescuing effects of α -ketoglutarate (energy source), asparagine (nitrogen source), and ROS scavengers (stress inhibitor) upon Gln-withdrawal. Based on the data, we only observed rescuing effects from α -ketoglutarate but not from asparagine and ROS scavengers, suggesting the importance of Gln as an energy source in supporting cells with dysregulated “Fbxo4-cyclin D” axis.

Line 158: Figure S6C does not have Gln positive control.

Reply: Per reviewer’s suggestion, we have re-run this blot with Gln positive controls; furthermore, we also included FACS to demonstrate cell apoptosis (Supplementary Fig. 10c&d).

Line 161: Figure 4E: Legend on this figure is not legible.

Reply: We re-labeled this figure.

Line 167: “Correlates” instead of “leads to”.

Reply: We substituted “leads to” with “correlate with”.

Line 172-174: Only metformin is used to treat type II diabetes, while phenformin is not FDA approved.

Reply: This is our mistake and we have changed this statement.

Line 174: What were concentrations of CB-839, Phenformin and Metformin used in Figures 5A-5I?

Reply: Per reviewer’s advice, we clearly indicated the concentrations of these compounds in the relative figure legends.

Line 175-179: Combination therapy does not appear to induce significantly increased “apoptotic rate” when Cyclin D (WT or T286A) is overexpressed – cleaved PARP is not different in lanes with single CB-839 compared to CB-839+Metformin (Figure 5D). Blots showing cleaved PARP in Figure S7 are underexposed thus making any interpretation very hard.

Reply: Good point. We have re-run the PARP blot and replaced the old one (Fig. 5d); meanwhile, we also run FACS to demonstrate the Annexin V-positive cells after ectopic cyclin D1 expression in TE15 cells (Supplementary Fig. 12). For blots in Fig. S7 (now it is Supplementary Fig. 11), we re-run the samples and involve PARP blots with longer exposure.

Line 224-225: From the Figure S12B, only TE7 cells showed increased cleaved PARP levels in response to Gln depletion (comparing TE7 Parental to PDR), but not TE10 cells. Please revise text to more clearly describe data shown in Figures.

Reply: Thank you. Due to the poor quality of these blots, we collected new samples and re-run

the blots (Supplementary Fig. 17c), which clearly demonstrate more PARP cleavage in both cell lines resisting to palbociclib.

Line 227: If it is impossible to establish resistant cells (resistant to what?) what type of cells were described on line 222-223?

Reply: In this statement, we mentioned “it is impossible to establish TE15 cells resistant to palbociclib”. We didn’t make the statement clear enough, but now we have added an additional clarification to describe what cells are being used for a specific experiment, and we also put in more details to characterize palbociclib resistant cells (Supplementary Fig. 18).

Reply to Reviewer #3:

In this study Qie et al. demonstrated that esophageal squamous cancer cells with dysregulated cyclin D1-CDK4/6 either due to genetic mutations or loss of Fbxo4, an E3 ubiquitin ligase for the cyclin complex present Glutamine-addiction. The authors further showed that dysregulation of Fbxo4-cyclin D1 axis leads to mitochondrial dysfunction and metabolic reprogramming. At the molecular level the authors found that Rb and mTORC1 contribute to Glutamine-addiction upon the dysregulation of the Fbxo4-cyclin D1 axis. Rb loss is accompanied by increased levels of glutaminase 1 (Gls1) which promotes Gln-addiction. More importantly the authors showed that combined treatment with CB-839, a GLS1 inhibitor and metformin/phenformin effectively induces cell apoptosis and suppresses cell proliferation in vitro, in xenografts and in palbociclib resistant tumors. These findings are novel and provide important insights into the ESCC vulnerability where cyclin D1-cdk4/6 abnormalities are common. The experiments were well designed and performed, and the manuscript was well written.

Reply: Thank you for your positive comments on our manuscript.

That said this reviewer has some concerns that require further experiments or discussion.

1. Validate changes in the levels of some putative downstream target of the mTOR pathway in ESCC cells at protein or transcript level after inhibitor treatment.

Reply: Thank you for this suggestion and we have tested downstream factors of mTORC1 in some blots in Fig. 7, for example, phosphorylation of S6 and 4EBP1. To further address reviewer’s concern, we have included more western blots to demonstrate the activity of mTORC1 pathway (Fig. 5 and Supplementary Fig. 11 & 17).

2. What is the level of Glis in ESCC biopsies?

Reply: In Supplementary Fig. 3d and 3e, we now demonstrate increased GLS1 mRNA levels in ESCC relative to normal tissues. Oncomine analysis was also performed to indicate that GLS1 is upregulated in human ESCC tumor tissues with statistical significance. Furthermore, independent analysis of another two GEO datasets also indicates GLS1 is elevated in ESCC comparing to normal tissues (Supplementary Fig. 4).

Reviewers' Comments:

Reviewer #2:

Remarks to the Author:

The authors provide revised manuscript titled "Targeting Glutamine-addiction and Overcoming CDK4/6 Inhibitor Resistance in Human Esophageal Squamous Cell Carcinoma". The authors addressed a number of reviewers' comments and have improved manuscript. The addition of the schematic in Figure 8f helps in clarifying proposed mechanism. However, mTOR data does not fit well with the rest of the manuscript and still leads to a disjointed paper. In addition, there are still a number of inconsistencies in the data that hamper this manuscript.

1) Supplemental Figure 1a: there doesn't appear to be much of a difference in cyclinD levels between Fbxo4^{+/+} and ^{-/-} MEFs. This is part of the main hypothesis of the paper, therefore it is not clear why are MEFs showing inconsistent levels of cyclinD.

2) Supplemental Figure 2a: Overexpressing D1 T286A in Fbxo4^{+/+} and ^{-/-} MEFs leads to very different levels of cPARP and CC3. This would suggest that additional pathways/proteins are perturbed in Fbxo4^{-/-} MEFs that contribute to increased signaling in the apoptosis. Are there any other targets of Fbxo4 that could contribute to the observed differences?

3) Supplemental Figure 6a: Alternative explanation for this data set, is that cMyc regulates a number of anti-apoptotic genes, consistent with cMyc knockdown leading to the increased cPARP levels. Additionally, levels of cMyc in TE7 and TE10 correlate with Gln uptake (S6a and S7a). It is clear that authors are trying to show that CyclinD is more relevant for Glutamine-addiction in the ESCC cell lines compared to cMyc, but this examination is more compelling in data shown in Sup Figure 17b, where cMyc knockdown TE7 and TE10 does not lead to loss of GLS.

4) CyclinD levels are also not consistent in Figure 7c and 7e. In figure 7c, PDR versions of TE7 and TE10 cell lines show increased cyclinD compared to parental TE7 and TE10. However, in Figure 7e, only TE10 shows increase in cyclinD levels in PDR version, but TE7 shows no increase in cyclinD levels in PDR line. This is inconsistent and suggests that there is a lot of variability in these cell lines; it is also concerning, as these are the main players in the proposed pathway.

Reviewer #3:

Remarks to the Author:

The revised manuscript has addressed this reviewer's concern.

Reviewer #4:

None

Reviewers' comments:

Reviewer #2 (Remarks to the Author):

The authors provided a revised manuscript titled “Targeting Glutamine-addiction and Overcoming CDK4/6 Inhibitor Resistance in Human Esophageal Squamous Cell Carcinoma”. The authors addressed a number of reviewers’ comments and have improved manuscript. The addition of the schematic in Figure 8f helps in clarifying proposed mechanism.

Reply: Thank you for your positive comments on our revised manuscript.

However, mTOR data does not fit well with the rest of the manuscript and still leads to a disjointed paper.

Reply: Thank you for mentioning this point. We respectfully, but partially disagree with the reviewer for this question. We think this question mainly attributes to the suppressing effects on mTORC1 signaling by metformin. In Figure 3, we found mTORC1 activation by cyclin D1-CDK can drive Gln-addiction. Theoretically, metformin should reduce the dependency on Gln through mTORC1 inhibition. **However, metformin is a compound with many different biological functions, there is a tendency to focus on one biological function at the expense of others.** In the context of our hypothesis, we took advantage of metformin as an inhibitor of mitochondrial respiration (OXPHOS). **Our data support the suppressing effects on OXPHOS overwhelm the rescuing effects related mTORC1 suppression, finally leading to cell apoptosis.** We undertook this line of investigation because our data suggest that cells with dysregulated cyclin D1-CDK demonstrate compromised mitochondria and they are dependent upon OXPHOS. Therefore, we suggest that the logic flow exists and it justifies the inclusion of the mTORC1 data. *In addition, there are still a number of inconsistencies in the data that hamper this manuscript.*

Reply: We have run new western blots to improve the data quality to address the reviewer’s concerns. Please see the new figures in the following questions/answers.

1) Supplemental Figure 1a: there doesn’t appear to be much of a difference in cyclinD levels between Fbxo4+/+ and -/- MEFs. This is part of the main hypothesis of the paper, therefore it is not clear why are MEFs showing inconsistent levels of cyclinD.

Reply: Thank you. To improve the data quality, we collected new samples and have re-run the western blot. Please find these new results in Supplementary Figure 1b.

2) Supplemental Figure 2a: Overexpressing D1 T286A in Fbxo4+/+ and -/- MEFs leads to very different levels of cPARP and CC3. This would suggest that additional pathways/proteins are perturbed in Fbxo4-/- MEFs that contribute to increased signaling in the apoptosis. Are there any other targets of Fbxo4 that could contribute to the observed differences?

Reply: Thank you. We agree with the reviewer that overexpressing D1 T286A results in different levels of PARP and caspase-3 cleavage, suggesting the possibility of existing other mechanisms. We have modified our interpretation by including one more sentence in Results and highlighted it in red. Our new statement is **“The above findings indicate cyclin D1, as one of the direct targets of Fbxo4, is required and sufficient to induce Gln-**

addiction.” We are screening for new targets, but this is a long and extensive process and we have no results to include in this regard currently.

3) *Supplemental Figure 6a: Alternative explanation for this data set, is that cMyc regulates a number of anti-apoptotic genes, consistent with cMyc knockdown leading to the increased cPARP levels.*

Reply: Thank you for your positive comments. We totally agree with the reviewer on this point but point out that in this case, knockdown of c-Myc does not reduce Gln-dependency supporting our conclusion that the mechanism of interest is independent of c-Myc overexpression.

Additionally, levels of cMyc in TE7 and TE10 correlate with Gln uptake (S6a and S7a).

Reply: We appreciate this question, but we cannot conclude that levels of c-Myc in TE7 and TE10 cells correlate with Gln uptake. Our data support that with increased cyclin D1 levels, TE7 and TE10 cells have more Gln uptake than TE15 cells; however, there is no direct support this increase is because of c-Myc. While in the classical way, c-Myc can upregulate Gln transporter and increase Gln uptake; however, besides c-Myc, many other factors can also regulate Gln uptake. For example, Gln transporters: ASCT2 can be regulated by E3 ubiquitin ligase RNF5 (Cancer Cell. 2015, 27(3): 354-69.). It remains elusive how Gln uptake is enhanced in ESCC cells.

It is clear that authors are trying to show that CyclinD is more relevant for Glutamine-addiction in the ESCC cell lines compared to cMyc, but this examination is more compelling in data shown in Sup Figure 17b, where cMyc knockdown TE7 and TE10 does not lead to loss of GLS.

Reply: We agree with the reviewer on this point. According to our data, ESCC cells, especially, the PDR cells, have a c-Myc-independent mechanism to induce GLS1 expression. Future study will dissect out the detailed mechanisms.

4) *CyclinD levels are also not consistent in Figure 7c and 7e. In figure 7c, PDR versions of TE7 and TE10 cell lines show increased cyclinD compared to parental TE7 and TE10. However, in Figure 7e, only TE10 shows increase in cyclinD levels in PDR version, but TE7 shows no increase in cyclinD levels in PDR line. This is inconsistent and suggests that there is a lot of variability in these cell lines; it is also concerning, as these are the main players in the proposed pathway.*

Reply: Thank you for the suggestion. We collected new samples and re-run blot for TE7 and TE7 PDR cells. Please see the new data in Figure 7e.

Reviewer #3 (Remarks to the Author):

The revised manuscript has addressed this reviewer's concern.

Reply: We appreciate your positive evaluation of our revised manuscript.

Reviewer#4:

Editorial note: Please find attached the comments of reviewer#4 to your rebuttal to Reviewer#1. You will see Reviewer#4 is mostly satisfied with your concerns and has only a few residual points to address:

Reply: Thank you for the positive comments about the improvement of our manuscript.

They raise the possibility that fig3e and 3b might have been mislabeled;

Reply: This is our mistake, we didn't present our rebuttal to this question clearly enough. Actually, we moved Panels a&b from Supplementary Figure 5 in old version (1st) to Figure 3 in revised version (2nd), therefore, we added two extra Panels and relabeled the new Figure 3 in revised version (2nd). Fig3e and 3b in the above question should refer to Fig 3g and 3d in the revised version (2nd). In the following, we include these two figures for clarification (highlighted in red boxes).

The above figure is Figure 3 from old version (1st).

The following Figure is Figure 3 from revised version (2nd).

They find that regarding the GSEA analysis, the statement that ESCC is a nutrient deprived cancer lacks evidence and this data does not add much to the paper;

Reply: We totally agree with the reviewer that without further solid biochemical data, only GSEA analyses cannot support that ESCC is a nutrient-deprived cancer. Indeed, in our manuscript, we don't emphasize glutamine is deprived in primary ESCC. **In addition, this idea grew out of a previous reviewer's question.** We used this published GSEA Gene Set to investigate whether there is an alteration of glutamine metabolism genes. The previous findings defined the Gene Set and named it "PENG_GLUTAMINE_DEPRIVATION_DN" (http://software.broadinstitute.org/gsea/msigdb/cards/PENG_GLUTAMINE_DEPRIVATION_DN.html).

To address the reviewer's concern, we have changed the figure legends for both Figure 2b and Supplementary Figure 3b in order to reduce the possible misleading interpretation of these data.

If the reviewer thinks this is not relevant, we can remove these data.

They find that the TE7 -Gln blot is not very convincing and should be exchanged.

Reply: Thank you. After examination of Figure 2 in this context, we found that this concern might be due to data presentation. As shown below and in the manuscript the data are precisely as predicted and consistent. In Figure 2d, we loaded our western blot samples in a sequence "without or with Gln", but in Figure 2h, we showed the flow cytometry data in an opposite sequence "with or without Gln" (we highlighted this in the following figure). Regardless of this, the data are totally supporting our hypothesis. We are happy to change the order of presentation if it is felt this would be of benefit to readers.

Reviewers' Comments:

Reviewer #2:

Remarks to the Author:

The authors have answered the majority of our concerns. However, a few concerns still remain. The authors did not answer our question of why CyclinD levels were inconsistent in Figure 7. They simply ran new western blots. A clear explanation would be helpful. The inclusion of data examining mTOR signaling still reads as tangential to the main story, which is focused on inhibiting GLS and OXPHOS in ESCC. This appears to be a point on which we differ. There were a number of sentences in the revised manuscript in which the authors' description of a noun was not preceded by the definite article. It is recommended the manuscript be thoroughly edited for grammar and clarity before publication.

REVIEWERS' COMMENTS:

Reviewer #2 (Remarks to the Author):

The authors have answered the majority of our concerns.

Reply: First, we appreciate the efforts of the reviewer to provide clear comments to help improve our manuscript.

However, a few concerns still remain.

Reply: We carefully discussed the following questions. Below, we present our explanation to address the reviewer's concerns.

The authors did not answer our question of why CyclinD levels were inconsistent in Figure 7. They simply ran new western blots. A clear explanation would be helpful.

Reply: Thank you for reminding us about the above question. We went through all our data records and found out that cyclin D1 levels are indeed upregulated when cells develop resistance to palbociclib; importantly, this upregulation is reproducible, and these findings are consistent with previous publications (please refer to the following papers). As for the cyclin D1 blot shown in previous version, we figured out that those lysate were collected from cells just after we thawed them from liquid nitrogen tank, and we think those cells might be compromised, and they proliferate poorly; however, in healthy and proliferating cells (at different passages), cyclin D1 upregulation is stable and reproducible in the PDR cells.

The mechanism of cyclin D1 upregulation remains elusive. One possibility is the presence of a feedback regulation: when CDK4/6 activity is suppressed, the cells upregulate cyclin D1 to antagonize the inhibiting effects of palbociclib. We remain intrigued by this observation and will continue to investigate the detailed mechanisms.

Paper list:

- 1. Oncogene. 2017 Apr 20;36(16):2255-2264. Figure 6c**
- 2. Oncotarget. 2018 Aug 3;9(60):31572-31589. Figures 1a, 3d, and 5f**

The inclusion of data examining mTOR signaling still reads as tangential to the main story, which is focused on inhibiting GLS and OXPHOS in ESCC. This appears to be a point on which we differ.

Reply: We agree with the reviewer that we do have different opinion on the necessity of involving mTORC1 data in this manuscript. We think this disagreement totally depends on the angle from which we perceive these data.

The following explains our logic why including these data:

First, after we found dysregulation of Fbxo4-cyclin D1 drives Gln-addiction, we focused on investigating downstream signaling pathways of cyclin D1 that drive Gln-addiction. According to our data, we found that besides Rb, mTORC1 signaling is also required to enhance cellular dependency on Gln. Therefore, the inclusion of mTORC1 data do have logical basis.

Second, mTORC1 is a central regulator to balance cell proliferation/growth and the availability of energy and nutrients. Hyperactivation of mTORC1 drives energy consumption, leading to energetic catastrophe, and finally cell apoptosis (Mol Cell. 2010 May 28;38(4):487-99.). Like the reviewer mentioned above, another merit of our manuscript focuses on developing new therapies for tumours with dysregulated Fbxo4-cyclin D1 axis. Before we moved onto this question, we investigated the underlying mechanisms why cells require Gln for survival. Our data support that Gln is indispensable for energy production in those cells. As a downstream factor of Fbxo4-cyclin D1, mTORC1 might drive Gln-addiction through enhancing energy consumption. This is the other reason why we present mTORC1 data. We have added a brief discussion of the above rationale in the Discussion part to make the inclusion of mTORC1 data reasonable.

There were a number of sentences in the revised manuscript in which the authors' description of a noun was not preceded by the definite article. It is recommended the manuscript be thoroughly edited for grammar and clarity before publication.

Reply: Thank you for your intensive reading of our revised manuscript. Following the reviewer's suggestion, we have gone through the whole manuscript, provided background information, and made changes wherever necessary.